# Education for Sustainable Development: Mapping the SDGs to University Curricula

**Thomas Adams** [1,2,*] , **Syed Muslim Jameel** [1] and **Jamie Goggins** [1,2,3,*]

1   School of Engineering, College of Science and Engineering, University of Galway, University Road, H91W2TY Galway, Ireland
2   MaREI, The Science Foundation Ireland Research Centre for Energy, Climate and Marine, Ryan Institute, University of Galway, University Road, H91W2TY Galway, Ireland
3   Construct Innovate, Ireland's National Construction Technology Centre, University of Galway, University Road, H91W2TY Galway, Ireland
*   Correspondence: t.adams4@universityofgalway.ie (T.A.); jamie.goggins@universityofgalway.ie (J.G.)

**Abstract:** Education for sustainable development (ESD) is a growing research field, particularly over the last decade. Measuring the level of ESD that is currently embedded in curricula is useful for planning the further implementation of sustainability-related teaching. The Sustainable Development Goals (SDGs) are a useful benchmark for sustainability topics and so this paper follows a methodology in which a keyword scanning tool was created to quantify the level of SDG coverage within a list of learning outcomes for a module. The aim of the research is to further develop this methodology and compare the results from the keyword tool with results from a survey of the academic staff who deliver the modules. SDG-related keyword lists were collected from multiple sources for a meta-analysis, examining the performance of various lists. These lists were then compiled into one list of over 12,000 SDG keywords and a team of reviewers conducted a critical analysis on the relevancy of the context in which the keywords were found when scanned. This process reduced the list to 222 "crucial keywords" and gave the keywords a relevancy label based on the STARS definitions, the sustainability tracking assessment and rating system. Finally, ChatGPT was also investigated as a method of enriching the critically analysed list with contextually relevant synonyms. A survey was carried out within the College of Science and Engineering at the University of Galway. It asked staff to rate the level of SDG coverage within their own modules, in their own opinion. This gave results which could be compared with the keyword scanning tool. The findings show success in improving the accuracy of the SDG keywords. ChatGPT added synonyms to the crucial keywords identified and this list was the most accurate out of all keyword lists used in the study. Using these keywords and the modules that staff rated in the survey, a correlation was found in the SDG trend.

**Keywords:** sustainable development goals; education for sustainable development; curriculum mapping; learning outcomes; pedagogy; sustainable education





## 1. Introduction

Education for sustainable development (ESD) is the education sector's response to the urgent and dramatic challenges that the planet faces, according to UNESCO [1]. As a field of research, ESD has been growing significantly ever since the United Nations declared 2005–2014 as the "Decade of Education for Sustainable Development" [2–8]. ESD has been around since long before this decade and the Sustainable Development Goals (SDGs) themselves. Lozano et al. [9] summarise a history of eleven global declarations, charters, and partnerships for sustainability in Higher Education institutions (HEIs), which goes back as far as 1990. ESD has its own target within the SDGs, under SDG 4: Quality Education. SDG target 4.7 promotes the worldwide adoption of ESD by 2030 [10].

The world has just passed the midpoint of the timeline for that target and, while ESD has grown significantly as a research field, it still has a long way to go with regards to

its adoption and implementation into education systems [11]. A prevalent point that is repeated in research is the importance of the holistic, whole-of-institution, and horizontal implementation of ESD for the best results [2–8]. Furthermore, the precept of "What gets measured gets done" alluding to the many sustainability assessment tools that have been created for HEIs to gain a baseline sustainability evaluation. The barriers to adopting an ESD policy and implementation plan include investment in ESD training for staff, the creation of a standardised ESD curriculum, and collaboration between institutions, organisations, and stakeholders on ESD [12]. The holistic implementation of ESD has been found to be the most effective route when it comes to incorporating sustainability teaching into curricula. To implement ESD holistically, one of the first steps needed is to measure the current level of ESD already embedded in each curriculum [13]. While ESD is a long-established field of research, the topic of mapping and monitoring ESD implementation is in its preliminary stages [14]. Understanding the embeddedness of sustainability in the curriculum can offer guidance on how to improve contextual approaches to ESD [15]. It can help to start a dialogue with module owners by highlighting modules that already embed certain SDGs into the curriculum, the level to which these SDGs are covered, and by showing which SDGs are not yet covered in the curriculum. It initiates the institution-wide process by scanning the curriculum institution wide [16].

Many schools, universities, and governments are attempting to align their education systems with the SDGs [17]. When it comes to aligning curricula with ESD, UNESCO [18], the Commonwealth [19], and the Sustainable Development Solutions Network (SDSN) Australia Pacific [20] have all released comprehensive guides for the incorporation of sustainability into education institutions. These guides and reports all agree that mapping the curriculum to the SDGs is useful as a baseline from which to build these implementation strategies. In Ireland, the government has been publishing national strategies on education for sustainable development since 2014, with the latest being released in 2022, the second national document on education for sustainable development [21]. This was released alongside a national ESD implementation plan for 2022–2026 [22]. This strategy is divided into five priority action areas, each with a set of objectives. The first action area is "Advancing Policy" and objective 1.4 under this area reads as follows: "Mapping and monitoring: Develop existing and new frameworks and tools to monitor and evaluate progress on ESD and enhance accountability". It references the use of ESD measurement tools to monitor implementation and progress towards SDG 4.7. The University of Galway's Sustainability Strategy, similarly to many other Universities, has the objective of embedding sustainability principles and practices into all programmes delivered at all levels of research, learning, and events [23]. I. D'Adamo and M. Gastaldi emphasise the importance of integrating sustainability into higher education, noting that "the majority of HEIs have not fully embraced sustainability in the curriculum" [24]. HEIs have two main methods of promoting sustainability, institutional initiatives and actual campus operations [25]. Implementing the SDGs into curricula is vital in utilising HEIs to their full potential in the sustainable transition.

In terms of the assessment of sustainability in curricula, there are very few free access tools available for use. A 2012 study conducted a review of 16 sustainability assessment tools and a 2019 study reviewed 19 sustainability assessment tools for HEIs, to examine trends in issues and methodologies [26,27]. These studies found that the tools focus on the environmental impacts of university operation and issues related to governance, lacking the measurement of curriculum, research, and outreach aspects. Further development of the assessment tools for the educational and outreach aspects of sustainability in HEIs is needed. As mentioned above, there are several guides available which offer advice to HEIs for implementing sustainability into education [18–20]. However, these guides do not provide a tool with which to assess the current level of education for sustainable development within a university's curriculum. This initial step of quantifying the level of ESD within a university allows for the identification of each institution's unique pathway for further ESD implementation, supporting the mantra of "what gets measured gets done" [28]. There has been more progress in measuring the level to which SDGs are embedded in research. A

machine-learned approach was used by Elsevier in the "Mapping Research Output to the SDGs" initiative [29–31]. These models use handpicked research publications that relate to an SDG to create a "golden set" for the machine-learning methodology. Elsevier created this for institutions to analyse and benchmark the impact of the institution's research on the SDGs [32]. This approach to categorising research papers under the SDGs is what this study suggests could be implemented for modules and programmes within HEIs.

This research expands on previous research which evaluated the level to which the SDGs were embedded within engineering degree programmes [33]. An output of this research was an Excel spreadsheet which could scan a list of learning outcomes for the presence of SDG-related keywords. The previous study used the list of 915 SDG related keywords compiled by researchers at Monash University [34]. This research paper aims to expand on Adams et al.'s research with three distinct aims. Firstly, to go beyond the Monash list of keywords and to analyse the performance of multiple different keyword lists and critically analyse these lists to identify crucial keywords. Secondly, to examine the use of ChatGPT as a method for enriching these crucial keywords lists with contextually relevant synonyms. Thirdly, to carry out a survey of module owners to gather qualitative and quantitative data to compare alongside the keyword scan results. This survey was adopted from a similar survey created by Philippe Lemarchand of the Technological University Dublin. From these aims the following research questions were composed:

RQ1: How accurate are various SDG keyword lists when scanned for in educational curricula, and can ChatGPT enhance the accuracy of keyword lists?

RQ2: How do keyword scans compare to surveys in terms of a method of ESD baseline measurement?

RQ3: What barriers and opportunities are there for staff in terms of ESD implementation?

This paper is placed in sustainability assessment in Higher Education institutions (HEIs) and focuses on curriculum assessment. It aims to address the acknowledged gap within sustainability assessment tools which is a lack of methodologies for assessing the education on sustainability topics [26]. The SDGs are used as a sustainability framework and this study's objective is to develop an automated approach, enhanced using artificial intelligence techniques, to create a baseline scan of the SDGs within curriculum material.

## 2. Materials and Methods

### 2.1. Relevancy of the Sustainability Tracking Assessment and Rating System (STARS)

STARS is a programme of AASHE, the Association for the Advancement of Sustainability in Higher Education [35]. STARS stands for the sustainability tracking assessment and rating system, a sustainability assessment tool for universities which has awarded a STARS rating to 588 institutions worldwide, as of March 2023 [36]. STARS, similar to many other HEIs' (Higher Education institutions') sustainability assessment platforms, categorises sustainability data into the following headings: Academics, Engagement, Operation, Planning & Administration, and Innovation & Leadership. Under each category there are subcategories which require data to be rated. One of the subcategories under 'Academics' requires institutions to report on the total number of courses or modules offered in an academic year, followed by the number of these that are sustainability-focused or sustainability-inclusive. To complete a STARS application, one must follow the STARS 2.2 Technical Manual [37], which includes a description of each subcategory and the data required. In this subcategory, the manual describes definitions for a module which could be considered sustainability-focused or sustainability-inclusive. The manual goes into further detail to give definitions at the learning outcome level for "sustainability-focused" or "sustainability-supportive" learning outcomes. These definitions were utilised in this study so that the SDG keyword analysis could become aligned to the criteria described in the STARS 2.2 Technical Manual. This method of aligning SDG scanning tools to STARS has been used in other recent studies [38]. Using these definitions gives the tool the potential to be used in STARS applications, where it is required to list modules which are

sustainability-focused or sustainability-supportive. The definitions for learning outcomes which are SDG-focused or SDG-supportive can be seen in Appendix A.

### 2.2. Keyword SDG Mapping Methodology

The first section of the methodology pursued in this study was to create a tool that could scan curriculum material for a list of SDG-related keywords, resulting in a baseline measure of SDG-related teaching in a university. The methodology of identifying meaning in large collections of text using keyword scans is long established [39], and it has been adopted to contextualise text aligned to the SDGs in recent years [40–43]. As mentioned, this study expands on previous research in which a tool was created using Microsoft Excel that crosschecks a list of learning outcomes for the presence of keywords [33]. This tool allows for the input of up to 3000 learning outcomes, which were then operated on by Equations (1) and (2). These equations scan the learning outcomes for the presence of SDG-related keywords, creating an output of learning outcomes now labelled with keywords and SDGs. Equation (1) operates by scanning a cell containing a learning outcome for the contents of another cell containing a keyword. If the keyword is present, the equation prints this keyword; if not, the cell remains blank. Equation (2) gathers the printed keywords. Equation (1) would then scan this cell for the keywords and print the SDG label. An open access version of the Microsoft Excel tool can be found in the Supplementary Material to this paper.

$$= if(isnumber(search \text{ “cell”},\text{“range”})),1,\text{“”}) \tag{1}$$

$$= textjoin(\text{“,”}, ,\text{”row of cells”}) \tag{2}$$

In the previous study, Microsoft Excel was used as the platform to perform the scan for ease of use. However, Python is much more capable of dealing with substantial amounts of data, so it became necessary to move the scan to this platform. The code created required two inputs—one spreadsheet containing curriculum material to be scanned and one containing SDG keywords. This gave the benefit of being able to easily test different keyword lists, something the Excel tool was less flexible at achieving. Python was also much faster, less computationally intense, and had no limit on the number of learning outcomes or keywords that could be inputted, while the Excel tool had limitations in these areas. A copy of the code used, and some explanatory pseudocode, can be seen in Appendix B. Figure 1 graphically demonstrates the processes the Python tool was programmed to go through.

In the flowchart, each box represents a step in the algorithm, and the arrows represent the flow of data between the steps. The boxes are labeled with brief descriptions of the steps they represent. Note that this algorithm assumes that there are functions named preprocess, match_words, and create_csv that are defined elsewhere in your program, and that you are passing in the necessary arguments to these functions. Additionally, you will need to define curriculum_metadata as a DataFrame containing the learning outcomes to match against.

Step 1: Reading data from Excel to DataFrame: In this step, we read the data from an Excel file and store it in a DataFrame object. A DataFrame is a two-dimensional table-like data structure that is commonly used for working with tabular data in Python. We can use the pandas data analytics library in Python to read in the Excel file using the read_excel function. We pass in the path to the file as an argument to this function. The resulting DataFrame will have a row for each record in the Excel file, with columns corresponding to the different fields in the file.

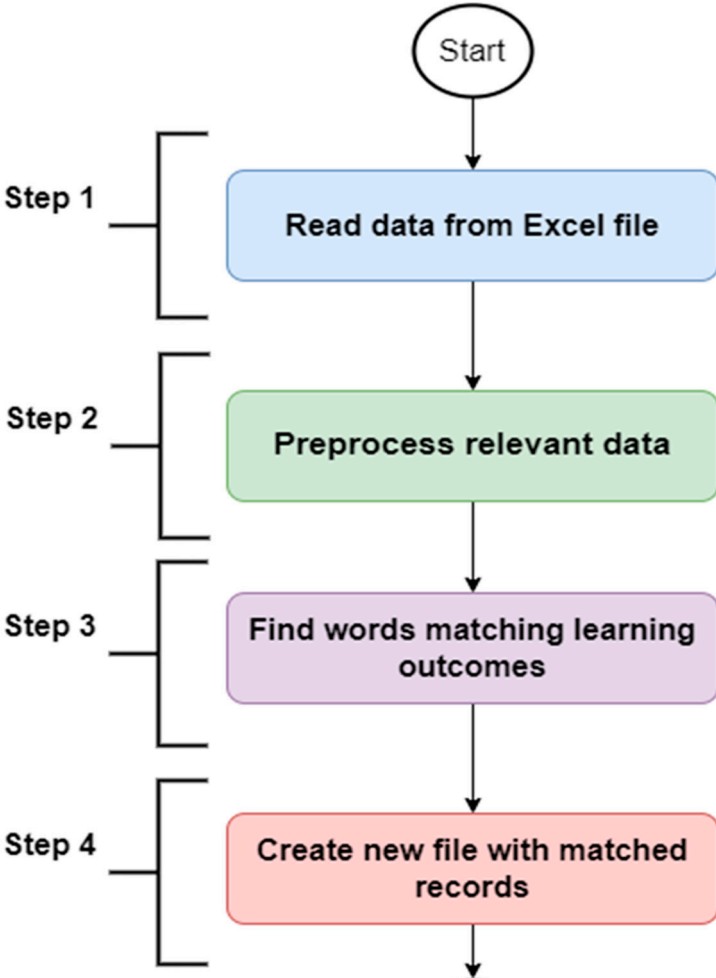

**Figure 1.** Steps followed in the keyword scanning automated tool.

Step 2: Preprocessing to extract relevant fields: In this step, we extract the relevant fields from the DataFrame and preprocess them to make them easier to work with. We begin by defining a list called records that will contain the preprocessed data for each record in the DataFrame. For each row in the DataFrame, we extract the values of the keywords (SDGall column), the SDG, and the relevancy columns, and preprocess them using a series of text-processing steps. These steps might include things such as converting the text to lowercase, removing punctuation, tokenising the text into individual words, and removing stop words. We then store the preprocessed values for each field in a dictionary called record and add this dictionary to the records list.

Step 3: Finding words matching learning outcomes: In this step, we compare the preprocessed text from Step 2 against a set of learning outcomes to see if there are any matches. We begin by defining an empty list called matched_records that will contain the records that match at least one learning outcome. For each record in the records list, we loop over the list of learning outcomes and check if any of the words in the learning outcome are present in the SDGall field of the record. We achieved this by defining a function called match_words that takes a piece of text and a list of words and returns 'True' if any of the words are present in the text. If we find a learning outcome that matches any of the words in the SDGall field, we set a flag called match_status to 'True'. After looping over all the learning outcomes, if match_status is 'True', we add the record to the matched_records list.

Step 4: Creating a new file with matched records: In this step, we create a new CSV file containing the data from the matched records. We define a function called create_csv that takes a list of records and a filename and writes the records to a CSV file with the given name. We pass the matched_records list and the filename 'all_sdgs.csv' to the create_csv

function to create the new file. Overall, this algorithm reads in data from an Excel file, preprocesses the relevant fields, finds records that match a set of learning outcomes, and creates a new file containing the matched records. The algorithm makes use of various text-processing techniques and the pandas data analytics library in Python to manipulate the data.

*2.3. Survey SDG Mapping Methodology*

This study utilised a survey to gather input from the academic staff (faculty) of modules through a survey which asked for staff to self-rate the modules they teach in terms of SDG coverage. The survey used in this paper was adopted from two sources. Firstly, a questionnaire was provided by Philippe Lemarchand of the Technological University Dublin, with the goal of gathering qualitative data related to the ESD to review alongside the quantitative data from the keyword scans. Secondly, an SDG Curriculum toolkit created by University College Cork (UCC) [44], which offers excel spreadsheets for students or staff to manually map modules, courses, or research to the SDGs. This was adopted into the survey and served as a second method of gathering quantitative data for a direct comparison with the keyword scan results. Academic staff (faculty) were asked to rate the coverage of the SDGs within their modules, with the added step of defining the level to which their modules covered each SDG—supportive or focused, according to the STARS definitions [37]. The survey was circulated among academic staff in the College of Science and Engineering at the University of Galway with responses received from 33 out of 217.5 FTEs (full-time equivalents), a response rate of 15.2%. The objectives of the survey were as follows:

- Gain an understanding of the level of interest in sustainability the respondents have.
- Collect suggestions for furthering sustainable development on campus.
- Inquire about what barriers surround ESD implementation.
- Collect suggestions for furthering ESD implementation in teaching and in research.

*2.4. Keyword Sources*

One of the research questions of this study was to identify how accurate various keyword lists are when scanned for in a set of curriculum material. This involved conducting a meta-analysis of the keyword lists in question. Five highly cited keyword lists were sourced online for this analysis, as summarised in Tables 1 and 2.

**Table 1.** Number of keywords that each list has for each SDG.

| SDG | Auckland | EU Mapper | Monash | Elsevier | SIRIS |
|---|---|---|---|---|---|
| 1 | 83 | 146 | 27 | 100 | 107 |
| 2 | 168 | 145 | 46 | 100 | 131 |
| 3 | 349 | 273 | 67 | 100 | 644 |
| 4 | 176 | 245 | 45 | 100 | 125 |
| 5 | 155 | 243 | 39 | 100 | 139 |
| 6 | 253 | 135 | 57 | 100 | 305 |
| 7 | 224 | 113 | 45 | 100 | 214 |
| 8 | 205 | 230 | 61 | 100 | 168 |
| 9 | 85 | 179 | 46 | 100 | 281 |
| 10 | 125 | 184 | 51 | 100 | 153 |
| 11 | 197 | 187 | 66 | 100 | 290 |
| 12 | 145 | 133 | 57 | 100 | 155 |
| 13 | 201 | 122 | 46 | 100 | 288 |
| 14 | 174 | 132 | 46 | 100 | 169 |
| 15 | 213 | 278 | 51 | 100 | 194 |
| 16 | 144 | 310 | 66 | 100 | 330 |
| 17 | | 326 | 31 | | |
| Misc | | | 68 | | |
| Total | 2897 | 3381 | 915 | 1600 | 3693 |

**Table 2.** Five keyword lists analysed in this study.

| Keyword List Source | Number of Keywords | SDGs Covered |
|---|---|---|
| Monash University [34] | 915 | SDG 1–17 [1] |
| Elsevier [31] | 1600 | SDG 1–16 |
| EU Mapper [45] | 3382 | SDG 1–17 (& targets) |
| SIRIS Academic [42] | 3693 | SDG 1–16 |
| University of Auckland [43] | 2897 | SDG 1–16 |

[1] Monash list also included a "Miscellaneous" category of keywords.

There are variations in the methodology followed for the creation of these keyword lists. The first keyword list was utilised in the creation of the Excel tool in the study prior to this paper. It is called the "Compiled Keywords for SDG Mapping", was created by Monash University in 2017, and contains 915 keywords spanning from SDGs 1 to 17 with an added miscellaneous category of keywords also [34]. It was created through a manual process by researchers at the University of Monash and SDSN Australia/Pacific. It is important to note that this list has not been peer-reviewed and it states that it is for guidance purposes only. The second keyword list is the Elsevier list of SDG search queries [31]. Since 2018, Elsevier has generated SDG search queries to help researchers and institutions track and demonstrate progress towards the targets of the United Nations Sustainable Development Goals (SDGs). The latest set of words claims a precision of over 80% when used for mapping research articles to the SDGs and it was created through multiple iterations of manual analysis and machine-learning approaches. The Elsevier words have a number weighting associated with each word, where the higher the number the more relevant the keyword to the SDG. These weights can be used to alter the SDG trend found in a scan. The third keyword list was created by researchers in the European Commission's Joint Research Centre. This list comes from a tool called the SDG Mapper, and this is the only list that goes further than the 17 SDGs and gives keywords for each of the 169 SDG targets [45]. Similar to the Monash list of words, the methodology behind the creation of this keyword list involved a manual textual analysis carried out by a group of experts [46]. The fourth keyword list was produced in 2019 by SIRIS Academic [42]. This list was built by extracting key terms from the UN's (United Nations') SDG indicators document, before being both manually and automatically enriched. A literature review around the SDGs informed the manual enrichment, and a machine-learning model based on neural networks was also utilised to automatically add to the database. This list contains 3445 keywords spanning SDGs 1–16. The fifth list used in this project was put together by the University of Auckland [43], which was based on Elsevier's SDG search query for published research. The University of Auckland used an n-gram text-mining model on abstracts of SDG-related academic publications to develop the list with additional search terms added thanks to input from the Sustainable Development Solutions Network (SDSN) and the UN [43]. This list contains 2321 keywords ranging from SDGs 1 to 16.

*2.5. Keyword Meta-Analysis*

A meta-analysis of these five keyword lists was conducted to gather quantitative results. A Python-based tool was used to perform the scan, in conjunction with a set of curriculum metadata in the form of 1617 modules, which represent a 45% sample of the active module list for the 2021/22 academic year at the University of Galway. Table 3 shows the spread of these modules across the colleges. A meta-analysis of different SDG keyword lists has not been published to date. Conducting this analysis also contributed to the next stage of the project, i.e., the critical analysis which would compile all keyword lists and then reduce the list down to a small, manageable size for a manual review. Conducting the meta-analysis first gave indications as to why certain lists perform well and why others do not. The objectives of the meta-analysis were:

- To identify the accuracy of each keyword list.
- To identify the frequency that keywords were hit.

- To review the level of SDGs in the curriculum for the 1617 modules.
- To investigate the correlation in SDG trends among various keyword lists.

**Table 3.** Curriculum metadata number of modules by college, University of Galway.

| College of | Number of Modules |
|---|---|
| Arts, Social Sciences & Celtic Studies | 576 |
| Science and Engineering | 568 |
| Medicine, Nursing, & Health Sciences | 228 |
| Business, Public Policy, & Law | 133 |
| Adult Learning and Professional Development | 112 |
| Total | 1617 |

The accuracy was calculated via an "accuracy factor". This was the total number of hits that a list achieved divided by the number of keywords in that list, giving hits per keyword or accuracy factor for each list. Analysing the frequency that keywords were hit allowed for the identification of "high, medium and low" frequency words, each to be analysed and processed in a separate way in the critical analysis. The general SDG trends among various lists can be compared to the SDG trend found after the critical analysis. It was also useful to identify modules and programmes with the most hits for the same comparison.

*2.6. Keyword Critical-Analysis*

The objective of the critical analysis was to identify the most effective or "crucial keywords" from the sample set. The five keyword lists were compiled into one list of over 12,486 keywords for critical analysis. This list was then input into the Python-based scanning tool and scanned in a set of curriculum metadata, consisting of 1617 modules and their learning outcomes. The critical analysis of the compiled keywords was conducted in five stages, as described in Figure 2.

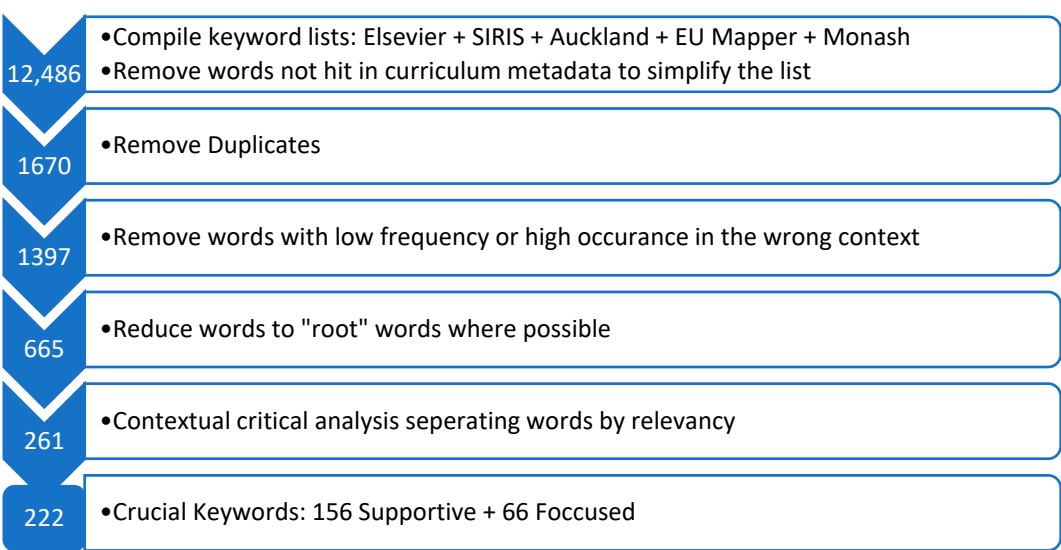

**Figure 2.** Stages in analysing the compiled list of SDG keywords.

In the first stage, words were removed if they were not hit in the scan of 1617 modules. The reason for this was to decrease the number of keywords to be analysed, as it was found that 87% of words did not receive any hit. The next stage was to compile the keywords that were hit in each list and remove duplicates. The third stage of the critical analysis involved removing words hit twice or less and reviewing words which were hit an abnormally large number of times. Words hit once or twice were not crucial to the analysis, accounting for 546 words removed. Of the high-hitting words, 186 were found to be in the wrong context.

An example of this was the keyword "age" in SDG 10, which was hit in the wrong context a huge number of times. The reason for this was because the scan operates by searching for an exact match of the keyword text-string, meaning "age" was hit in "management", etc. The fourth step involved removing all words for which a shorter root word was present. Using "Biodiversity" as an example again, the Monash list contains the following four terms which were all removed under this process "Coastal biodiversity, Biodiversity loss, Marine biodiversity, Strategic plan for biodiversity". These longer keywords are not necessary and will end up giving more than one hit if longer text-strings are present. Table 4 shows the frequency that keywords were hit, segregated into low, medium, and high hitters and outliers. Low-hitting words were deemed not crucial and were removed. The outliers and high-hitting words were reviewed for their context. Table 5 shows the top 5 keywords in terms of number of hits overall, their associated lists and SDGs, and the number of times that word was hit.

**Table 4.** Frequency in which keywords were hit.

| Category | Frequency | No. Words |
|---|---|---|
| Low-hitting words | 1–2 times | 546 |
| Medium-hitting words | 3–10 times | 417 |
| High-hitting words | 11–49 times | 275 |
| Outliers | 50+ times | 159 |
| Total | | 1397 |

**Table 5.** Top 5 keywords hit.

| Keyword | List | No. Times Hit | SDG |
|---|---|---|---|
| sea | Elsevier, SIRIS | 2150 | 13 & 14 |
| work | Monash, SIRIS | 1245 | 8 |
| age | Monash, SIRIS | 1092 | 10 |
| health | Auck, Monash, Elsevier, SIRIS | 1015 | 1, 3, & 10 |
| management | Auck, Elsevier, SIRIS | 955 | 6, 9, 14, & 16 |
| environment | Monash, Elsevier, SIRIS | 895 | 1, 2, 12, 13, & Misc. |

The final stage was a manual, in-depth contextual review, carried out on the remaining keywords to evaluate each word's relevancy. This analysis was performed by members of the university's Sustainability Academic Working Group. The team consisted of seven sustainability experts and each expert reviewed a list of learning outcomes and judged if the keyword found in the learning outcome was in the right context. There was cross-over in the learning outcomes reviewed by each expert to avoid bias entering the process. This critical review is where the STARS definitions for sustainability-focused or sustainability-supportive learning outcomes were applied. Once the reviewer understood the context that the keyword was in, a relevancy score was given to the keyword for each learning outcome reviewed. The relevancy scores, as can be seen in Table 6, were on a scale of 0 to 4.

**Table 6.** Description of Relevance Scores.

| Score | Relevancy Description |
|---|---|
| 0 | Not relevant to assigned SDG |
| 1 | Supportive of an SDG other than the SDG assigned |
| 2 | Focused on an SDG other than the SDG assigned |
| 3 | Supportive of assigned SDG |
| 4 | Focused on assigned SDG |

Zero meant the keyword was in the incorrect context in this case and should not be associated with the SDG. A score of 1 or 2 aligned to supportive or focused contexts, respectively, but these scores were assigned when a keyword was found in the context of a different SDG than the one it was currently assigned. A score of 3 or 4 meant the keyword was in the correct context in relation to the associated SDG. Assigning a score of 3 meant the keyword was in a context that was supportive of that SDG, as defined in the STARS 2.2 Technical Manual [37]. A score of 4 meant the keyword was in a focused context as defined by STARS. After all learning outcomes for each keyword had been analysed, the relevancy scores were summed to see if the keyword was found mostly in a focused, supportive, or irrelevant context. If the keyword received the same relevancy score for at least 70% of the learning outcomes, it would be assigned this relevancy label. If the review group did not reach a consensus on the relevancy label for a given keyword, the lowest of the scores for that keyword is assigned, i.e., if it received 50% "3—supportive" and 50% "0—irrelevant", it would be deemed irrelevant, if it received 60% "4—focused" and 40% "3—supportive", it would still be deemed supportive. Words labelled irrelevant were removed, and words labelled with scores 1 or 2 were moved to the more relevant SDG category. This critical analysis produced two final lists, that is, one with 156 SDG-supportive keywords and one with 66 SDG-focused keywords, giving a total of 222 keywords.

### 2.7. Keyword List Enrichment Utilising Artificial Intelligence

One of the project's objectives was to incorporate some aspects of machine-learning or artificial intelligence into the keyword scanning tool. ChatGPT was launched during the methodology phase of this study. Given its free access nature and its ability to understand the context of text, it was selected as the platform of choice. ChatGPT is a chatbot, developed by OpenAI, which has been built on large language models and was launched in November 2022 [47]. It is unique in that it is freely available to use, and it interacts in a conversational way, while also being capable of articulating expert-level answers across many fields. It is important to note that it is not perfect and can generate factually incorrect material, so everything should be checked for correctness [48].

The critical analysis described in Section 2.7 produced a concise list of 222 keywords reviewed for accuracy and labelled with a relevancy that aligns with STARS. The aim was to capture the progress made in this critical analysis and enhance it further by adding synonyms of the keywords to the list. Without ChatGPT, this would be a challenge given there were 214 words and any synonyms added to the list had to be in the context of the SDGs, to avoid going backwards in the critical analysis. A previously considered method was to search for synonyms in a large lexical database of English, such as WordNet [49]. The issue here was that many of the synonyms produced were not relevant to the SDGs.

ChatGPT can understand context, so it was possible to simply ask ChatGPT the following: "Can you provide me with an extensive list of synonyms for the following keywords in the context of SDG X". Table 7 shows an example of three synonyms that ChatGPT produced for the keyword Solar under SDG 7. This method produced an average of 4.9 synonyms per keyword, increasing the keyword list from 222 to 1303. The list of synonyms was cross-checked with the list of keywords that were removed during the critical analysis to ensure that this step did not add any words back in that had been removed earlier. This step removed 118 words from the synonym list, meaning the final GPT list comprised the 222 crucial words plus 963 synonyms, which is a total of 1185 words.

**Table 7.** Sample Synonym List.

| SDG | Keyword | Synonyms |
|---|---|---|
| 7 | Solar | Photovoltaic |
| | | Sun-powered |
| | | Insolation |

## 3. Results

### 3.1. Meta-Analysis

This section presents the results from the meta-analysis of five highly cited SDG keyword lists. Table 1 in Section 2.5 gives the number of keywords in each list. This table was used for calculations in the results. The objectives of the meta-analysis were:

- To identify the accuracy of each keyword list.
- To identify the frequency that keywords were hit.
- To review the level of SDGs in the curriculum for the 1617 modules.
- To investigate the correlation in SDG trends among various keyword lists.

A comparison of the trend across the SDGs can be seen in Figure 3. The trends in comparison are for the following four metrics: the total number of hits, the total number of keywords in the list, the number of modules hit, and the number of unique keywords hit. The graph shows the grand total for each of these metrics across all 12,486 keywords. Given such a high number of keywords, it is highly likely that the SDG trend from the hits will follow the trend from the number of keywords in the list to begin with. This was verified by the correlation coefficients between these datasets. A moderately strong positive correlation was found between the number of words under each SDG and the number of unique keywords hit with r = 0.76. This was similar for the number of modules hit and the number of keywords r = 0.74. This shows the effect of having too many keywords. A high number of keywords to begin with can skew the results by achieving too many hits. The fact that the overall SDG trend has a moderately strong correlation to the number of keywords shows that more words will inevitably give more hits, so there must be a useful limit on the size of keyword lists. Table 8 shows how this effect trickles down at the individual list level. The R-value here shows the correlation between the number of unique keywords hit for each list and the total number of keywords under each SDG for the compiled 12,486 keyword list. It can be seen that the lists with higher numbers of keywords have a higher correlation with the SDG trend from the total number of keywords. Elsevier and Monash have very weak negative and very weak positive correlations, respectively, and both lists have significantly fewer keywords than the other three lists, each of which has a strong positive correlation. SDG 17 is omitted from the results in Figure 3 as it received so few hits that it was not observable on the graph.

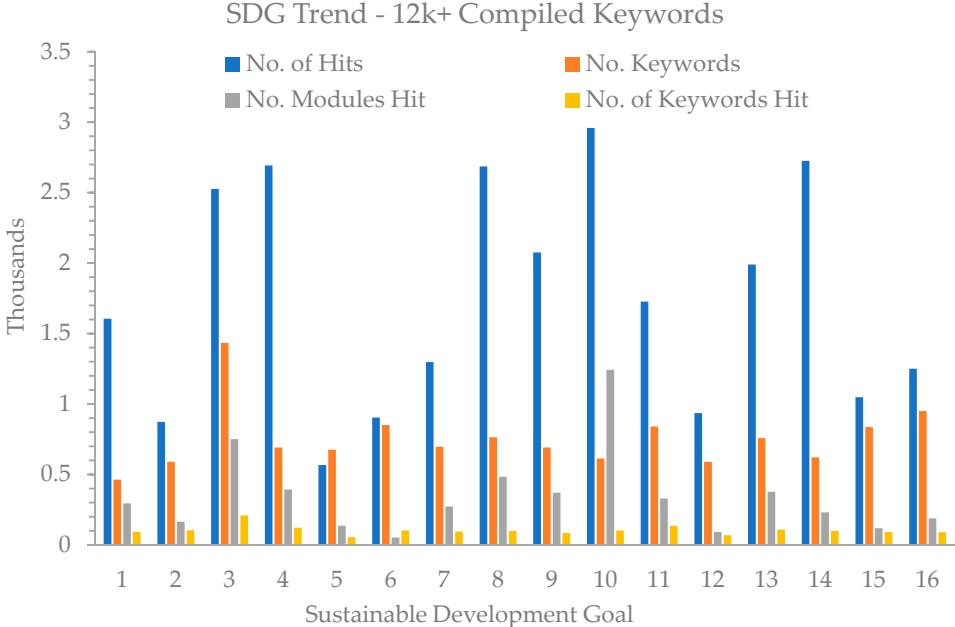

**Figure 3.** Overall SDG trend for the total 12,486 keywords, showing various metrics.

**Table 8.** Number of unique keywords hit from each list and correlation values between each list and the compiled list of 12,456 keywords.

| List | Auckland | EU Mapper | Monash | Elsevier | SIRIS |
|---|---|---|---|---|---|
| Total keywords | 2897 | 3381 | 915 | 1600 | 3693 |
| Unique keywords hit | 322 | 121 | 228 | 577 | 422 |
| R-value | 0.81 | 0.74 | 0.15 | −0.11 | 0.60 |

Figure 4 shows a comparison of each keyword list for the three metrics to be discussed in this section: total number of hits (frequency), the number of modules hit (sustainability percentage), and the number of unique keywords hit (accuracy). The accuracy of each keyword list was calculated by finding the percentage of words that were hit for each list. This was calculated by dividing the number of unique keywords that were hit by the total number of keywords in that list.

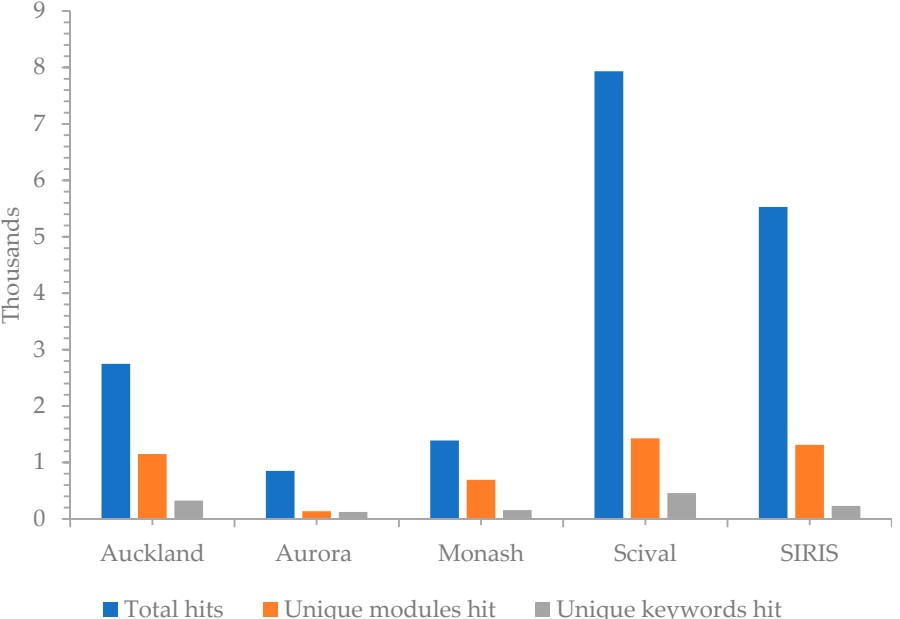

**Figure 4.** Totals of various metrics for each keyword list.

These data are summarised in Table 9. For example, 36% of the Elsevier list of keywords received hits. This is the highest percentage that any list received and so it is effectively the most accurate list as, proportionally, more of its words were hit than other lists. On average, 13% of words were hit or, in other words, 1670 of the 12,456 keywords were hit. As described in Section 2.7, after duplicates were removed this totaled 1397 unique keywords that were hit. SDG 1 keywords received the most hits, with 20% of SDG 1 words hit on average, and SDG 17 the least with 2%. Table 10 shows the number of unique modules hit for each keyword list divided by 1617, the total number of modules. This represents the percentage of sustainability that each list identified in the curriculum. This is an important figure to consider for accuracy as this shows the actual portion of the university's curriculum that each list has identified as related to an SDG. There is a lot of variation in these results, showing the inconsistencies that can occur due to the selection of keyword lists. The overall average shows that these lists consider 68% of the 1617 modules to be related to an SDG. This is exceedingly high considering that the average institution on the STARS system had just 13.3% sustainability modules, as seen on the STARS content display [50]. SDG 17 is omitted from the results Table 10 as it received zero hits.

**Table 9.** Percent of keywords that were hit for each SDG for each keyword list *.

| SDG | Auckland | EU Mapper | Monash | Elsevier | SIRIS | Average |
|---|---|---|---|---|---|---|
| 1 | 2% | 3% | 48% | 52% | 20% | 20% |
| 2 | 14% | 6% | 30% | 34% | 17% | 17% |
| 3 | 23% | 8% | 22% | 42% | 7% | 15% |
| 4 | 13% | 6% | 27% | 53% | 15% | 18% |
| 5 | 1% | 0% | 28% | 37% | 2% | 8% |
| 6 | 8% | 4% | 23% | 38% | 8% | 12% |
| 7 | 9% | 6% | 36% | 34% | 8% | 14% |
| 8 | 8% | 3% | 30% | 29% | 17% | 13% |
| 9 | 1% | 2% | 13% | 37% | 13% | 12% |
| 10 | 11% | 2% | 35% | 31% | 24% | 17% |
| 11 | 15% | 4% | 30% | 33% | 16% | 16% |
| 12 | 8% | 4% | 21% | 28% | 7% | 12% |
| 13 | 12% | 5% | 24% | 42% | 9% | 14% |
| 14 | 11% | 4% | 28% | 38% | 15% | 16% |
| 15 | 12% | 3% | 22% | 25% | 11% | 11% |
| 16 | 5% | 2% | 20% | 24% | 12% | 9% |
| 17 | | 1% | 10% | | | 2% |
| Misc | | | 13% | | | 13% |
| Total | 11% | 4% | 25% | 36% | 11% | 13% |

* This table is a heat map, where the darker shades of colour align with high table values.

**Table 10.** Percent of curriculum that contains keywords for each SDG for each keyword list *.

| SDG | Auckland | EU Mapper | Monash | Elsevier | SIRIS | Average |
|---|---|---|---|---|---|---|
| 1 | 0.1% | 0.1% | 7% | 11% | 0.2% | 4% |
| 2 | 1% | 0.1% | 3% | 5% | 1% | 2% |
| 3 | 29% | 0.1% | 5% | 6% | 7% | 9% |
| 4 | 8% | 0.3% | 2% | 12% | 3% | 5% |
| 5 | 1% | 0% | 1% | 7% | 0% | 2% |
| 6 | 2% | 0% | 0.4% | 1% | 0.2% | 1% |
| 7 | 2% | 0% | 2% | 10% | 2% | 3% |
| 8 | 12% | 0% | 5% | 10% | 2% | 6% |
| 9 | 3% | 7.2% | 8% | 3% | 1% | 5% |
| 10 | 1% | 0.1% | 36% | 1% | 39% | 15% |
| 11 | 5% | 0% | 4% | 3% | 8% | 4% |
| 12 | 0.2% | 0% | 1% | 4% | 1% | 1% |
| 13 | 3% | 0.1% | 0.4% | 14% | 6% | 5% |
| 14 | 2% | 0% | 1% | 2% | 9% | 3% |
| 15 | 2% | 0% | 1% | 2% | 2% | 1% |
| 16 | 0.2% | 0.1% | 1% | 3% | 7% | 2% |
| Misc | | | 3% | | | 3% |
| Total | 71% | 8% | 81% | 94% | 88% | 68% |

* This table is a heat map, where the darker shades of colour align with high table values.

Even though the Elsevier set of words performed well in terms of the percent of keywords hit, we can see the effect when such a high frequency of keywords are hit. According to the Elsevier set of keywords, 94% of the 1617 modules are related to an SDG. In fact, all the lists, bar one, marked over 70% of the curriculum as SDG-related. The EU Mapper list performed poorly in terms of accuracy or percent of keywords hit in comparison to the other lists, but it may show the closest real representation of sustainability in the

curriculum, as this list labelled just 8% of modules as associated with an SDG. The frequency of each keyword list is summarised in Table 11, which shows the total number of hits that each keyword list achieved divided by the number of keywords in that list. This differs from the percentage of keywords hit shown in Table 9, because the frequency considers if a word was hit more than once, counting the total hits for each list, as opposed to the total keywords. This is a frequency factor, showing the number of times a set of keywords received hits per keyword. For example, taking keywords for SDG 10 from Monash, we can see that there is a high chance of this set receiving hits. This result means the set received 22 times more hits than the number of words in the set, a highly frequent set of words. On average, there were 2.3 times more hits than keywords. This is high given there were 12,486 keywords, meaning there were 28,377 hits among the 1617 modules. Table 11 was useful for identifying outlier keywords that received an abnormally enormous number of hits, usually in the wrong context. High frequencies often correlated with these high hitting keywords, such as the keyword "age" in the Monash SDG 10 group, as discussed in Section 2.7. Similar to the results in Table 9, the Elsevier list had the highest frequency of hits followed closely by Monash, and the EU Mapper list received the lowest score.

**Table 11.** Frequency factor for each SDG for each keyword list *.

| SDG | Auckland | EU Mapper | Monash | Elsevier | SIRIS | Average |
|---|---|---|---|---|---|---|
| 1 | 0.1 | 0.04 | 14.6 | 11.2 | 0.7 | 3.5 |
| 2 | 0.4 | 0.14 | 8.8 | 2.5 | 1.1 | 1.5 |
| 3 | 2.6 | 0.32 | 4.6 | 4.7 | 1.1 | 1.8 |
| 4 | 2.6 | 0.18 | 3.0 | 18.5 | 1.7 | 3.9 |
| 5 | 0.1 | 0.004 | 2.1 | 4.5 | 0.2 | 0.8 |
| 6 | 0.4 | 0.10 | 1.7 | 3.7 | 1.1 | 1.1 |
| 7 | 0.4 | 0.11 | 2.8 | 9.2 | 0.7 | 1.9 |
| 8 | 2.0 | 0.06 | 10.5 | 8.7 | 4.4 | 3.5 |
| 9 | 2.2 | 2.99 | 12.5 | 6.0 | 0.6 | 3.0 |
| 10 | 0.4 | 0.02 | 22.0 | 5.7 | 7.9 | 4.8 |
| 11 | 0.7 | 0.04 | 5.7 | 6.5 | 1.9 | 2.1 |
| 12 | 0.2 | 0.05 | 4.4 | 5.9 | 0.4 | 1.6 |
| 13 | 0.5 | 0.30 | 2.7 | 10.8 | 2.2 | 2.6 |
| 14 | 0.6 | 0.06 | 2.5 | 8.9 | 9.5 | 4.4 |
| 15 | 0.4 | 0.05 | 2.5 | 4.8 | 1.8 | 1.3 |
| 16 | 0.1 | 0.05 | 2.2 | 2.9 | 2.4 | 1.3 |
| 17 | | 0.08 | 0.1 | | | 0.1 |
| Misc | | | 7.2 | | | 7.2 |
| Total | 0.9 | 0.3 | 6.0 | 7.2 | 2.1 | 2.3 |

* This table is a heat map, where the darker shades of colour align with high table values.

Tables 12–14 highlight findings from the top performing modules from the meta-analysis. Table 12 shows the 5 modules that received the most hits from each list, the darkness of the green tint represents the number of hits that the module received. Table 13 shows the seven modules from Table 12 which were one of the top five modules for at least two keyword lists, also showing the module title for context. Table 14 presents the number of hits, the number of unique keywords, and the SDGs covered across all 12,000+ keywords for the aforementioned seven modules.

**Table 12.** Module codes for five modules with most hits for each list, tint showing number of hits *,1.

| Auck | Eu Mapper | Monash | Elsevier | SIRIS | Crucial | GPT |
|---|---|---|---|---|---|---|
| CE3105 | MD201 | PAB5104 | PAB5104 | PAB5104 | PAB5105 | AY590 |
| CE6118 | ZO417 | PAB5105 | ZO417 | ZO417 | PAB5104 | CE514 |
| MD302 | MD304 | CE6103 | PAB5105 | OY2110 | ZO417 | PAB5105 |
| MD201 | PAB5104 | CE6118 | EG5101 | EC388 | CE343 | CE343 |
| ZO417 | CE6118 | CE3105 | EG400 | CE3105 | CE514 | MD201 |

\* This table is a heat map, where the darker shades of colour align with high table values. [1] Module information can be seen on University of Galway website [51].

**Table 13.** Seven modules which have been hit by more than one list in top ten modules.

| Module Code | Module Title | No. Lists Agree |
|---|---|---|
| PAB5104 | Gender, Agriculture & Climate Change | 5 |
| ZO417 | Marine & Coastal Ecology | 5 |
| PAB5105 | Low-Emissions & Climate-Smart Agriculture & Agri-Food Systems | 4 |
| CE3105 | Environmental Engineering | 3 |
| MD201 | Health & Disease | 3 |
| CE514 | Transportation Systems and Infrastructure II | 2 |
| CE343 | Sustainable Energy | 2 |

**Table 14.** Seven modules which have been hit by more than one list in top ten modules.

| Module | PAB5104 | ZO417 | PAB5105 | CE3105 | MD201 | CE514 | CE343 |
|---|---|---|---|---|---|---|---|
| No. Hits | 161 | 156 | 155 | 140 | 111 | 137 | 112 |
| No. Keywords | 61 | 53 | 63 | 59 | 55 | 47 | 44 |
| No. SDGs hit | 17 | 15 | 14 | 16 | 15 | 15 | 14 |
| Unhit SDGs | NA | 5/17 | 5/14/17 | 5 | 7/17 | 5/17 | 5/16/17 |

### 3.2. Critical Analysis and Chat GPT Enrichment

The critical analysis aimed to identify a list of "crucial keywords" or the words that are the most effective from the five lists analysed in the meta-analysis. The critical analysis produced a list of 222 crucial keywords which were split into 156 SDG-supportive words and 66 SDG-focused words, as per the STARS definitions in Appendix A. This list was then automatically enriched using the artificial intelligence platform ChatGPT, as described in Section 2.7. This section presents results showing the same metrics explored in Section 3.1, but for the "crucial" and "GPT" keyword lists. The average for the five previously published keyword lists (i.e., "Average" columns from Tables 9 and 11) have been incorporated into Table 15. And the same column from Table 10 into Table 16, to compare these critically analysed keyword lists with the average from the un-analysed lists that had been published previously. In Table 15, the improvement in the accuracy of the list after the critical analysis is clear. Elsevier showed the highest percentage of keywords hit at 36% in Table 9 and, when all lists were totaled, only 13% of their words received hits overall. The crucial keyword list shows the highest keyword coverage by far with 69%. This shows the team of reviewers who carried out the critical analysis successfully identified highly relevant words to the University of Galway's curriculum. This could show that these words are now biased towards the University of Galway. However, it is likely that the fact that the list only has 222 words in total had an effect. Fewer words in the list means fewer words are hit to score a high accuracy percentage. In the case of the crucial list, 153 words were hit. The GPT list shows positive results in Table 15 also, with an average accuracy equal to that of the Elsevier list, and higher than all other lists analysed in this study. SDG 17 is omitted from the results Tables 15 and 16 as it received zero hits.

**Table 15.** Accuracy and frequency results for critically analysed keyword lists ("Crucial"), critically analysed keyword lists enriched by ChatGPT ("GPT"), and average for the five previously published keyword list ("Average") *.

| SDG | Accuracy: Percent of Keywords Hit | | | Frequency: Hits/Words | | |
|---|---|---|---|---|---|---|
| | Crucial | GPT | Average | Crucial | GPT | Average |
| 1 | 43% | 16% | 20% | 1.3 | 0.9 | 3.5 |
| 2 | 100% | 27% | 17% | 10.3 | 0.7 | 1.5 |
| 3 | 70% | 33% | 15% | 4.9 | 3.1 | 1.8 |
| 4 | 82% | 43% | 18% | 18.8 | 8.3 | 3.9 |
| 5 | 33% | 50% | 8% | 5.3 | 7.9 | 0.8 |
| 6 | 100% | 43% | 12% | 9.3 | 2.0 | 1.1 |
| 7 | 80% | 50% | 14% | 5.9 | 9.6 | 1.9 |
| 8 | 71% | 59% | 13% | 5.4 | 5.6 | 3.5 |
| 9 | 89% | 58% | 12% | 27.4 | 13.9 | 3.0 |
| 10 | 73% | 43% | 17% | 3.6 | 7.7 | 4.8 |
| 11 | 60% | 27% | 16% | 4.0 | 8.0 | 2.1 |
| 12 | 70% | 35% | 12% | 4.0 | 2.0 | 1.6 |
| 13 | 72% | 28% | 14% | 4.2 | 4.2 | 2.6 |
| 14 | 67% | 24% | 16% | 7.2 | 9.2 | 4.4 |
| 15 | 56% | 36% | 11% | 5.3 | 5.6 | 1.3 |
| 16 | 57% | 30% | 9% | 2.9 | 4.4 | 1.3 |
| Total | 69% | 36% | 13% | 6.3 | 5.4 | 2.3 |

* This table is a heat map, where the darker shades of colour align with high table values.

**Table 16.** Percent of curriculum that contains keywords for critically analysed word lists ("Crucial"), critically analysed keyword lists enriched by ChatGPT ("GPT"), and average for the five previously published keyword list ("Average") *.

| SDG | Focused Hits | | Supportive Hits | | Average |
|---|---|---|---|---|---|
| | Crucial | GPT | Crucial | GPT | 5 List |
| 1 | 0.1% | 1.6% | 0.1% | 0% | 4% |
| 2 | 0% | 0% | 1.6% | 0.8% | 2% |
| 3 | 1.4% | 4.9% | 7% | 20.8% | 9% |
| 4 | 2% | 7% | 6% | 11.9% | 5% |
| 5 | 0.0% | 0% | 0.4% | 1.8% | 2% |
| 6 | 0.5% | 0.7% | 0.2% | 0.4% | 1% |
| 7 | 0.2% | 2.4% | 1.0% | 5.1% | 3% |
| 8 | 0% | 0% | 2.4% | 5.9% | 6% |
| 9 | 0% | 0% | 9% | 6.7% | 5% |
| 10 | 0.1% | 0.6% | 1.4% | 2.5% | 15% |
| 11 | 0.1% | 0.5% | 1.5% | 5.4% | 4% |
| 12 | 0.5% | 0.0% | 1.2% | 0.0% | 1% |
| 13 | 1% | 1% | 1.1% | 0.0% | 5% |
| 14 | 0.4% | 0% | 0.9% | 0.1% | 3% |
| 15 | 0.0% | 0% | 1.1% | 1.1% | 1% |
| 16 | 0.4% | 0.1% | 1.4% | 1.5% | 2% |
| Total | 7% | 19% | 36% | 65% | 68% |

* This table is a heat map, where the darker shades of colour align with high table values.

This list has 1185 words, meaning 465 words were hit. Again, this is bested only by the Elsevier list which had 576 words hit. The drop in accuracy from the crucial list to the GPT is due to it being a longer list of words. It is also worth noting that, as the number of words in a list goes up, this accuracy factor will most likely decrease, and so, given the Elsevier list has 1600 words compared to 1185 in the GPT list, it is technically more accurate than the GPT list. Looking at the right-hand side of Table 15, the frequency at which words were hit is more evenly spread in the GPT list than in the crucial list. There are a few outliers such as SDG 9 and 4 under the crucial list. These lists had an average frequency which was over 2.5 times higher than the un-analysed lists in Table 11. Table 16 presents the performance of the critically analysed and enriched lists as indicators of the level that SDGs are embedded in the curriculum. The benefit of the STARS relevancy labels can be seen straight away, with much more realistic percentages of focused modules shown at a total of 7% for the crucial list and 19% for the GPT. The level of supportive modules is much higher with totals of 36% for the crucial list and 65% for the GPT. However, these are still lower than the 68% average from the un-analysed list, showing the critically analysed and enriched lists are capturing a more realistic picture of where sustainability lies in the curriculum.

*3.3. Survey Questionnaire*

The objectives of the questionnaire section of the survey were as follows:

- Gain an understanding of the level of interest in sustainability the respondents have.
- Collect suggestions for furthering sustainable development on campus.
- Inquire about what barriers surround ESD implementation.
- Collect suggestions for furthering ESD implementation in teaching and in research.

The respondents had a high level of interest in sustainability. A total of 91% considered their work relevant to one of the pillars of sustainability—the environment, economy, or society. All thought their work to be relevant in helping society move towards a more sustainable future, with 65% claiming very relevant, 32% saying somewhat relevant, and 3% saying slightly. When asked if they had interest in joining a sustainability teaching group, 70% said yes. There were also no disagreements about developing the sustainability literacy of students, with 60% strongly agreeing that the university should be developing the sustainability literacy of its students, and the other 40% also agreeing. The survey asked staff to describe any opportunities they saw within their current role to help the university become a more sustainable institution. Figure 5 shows a word cloud of the results.

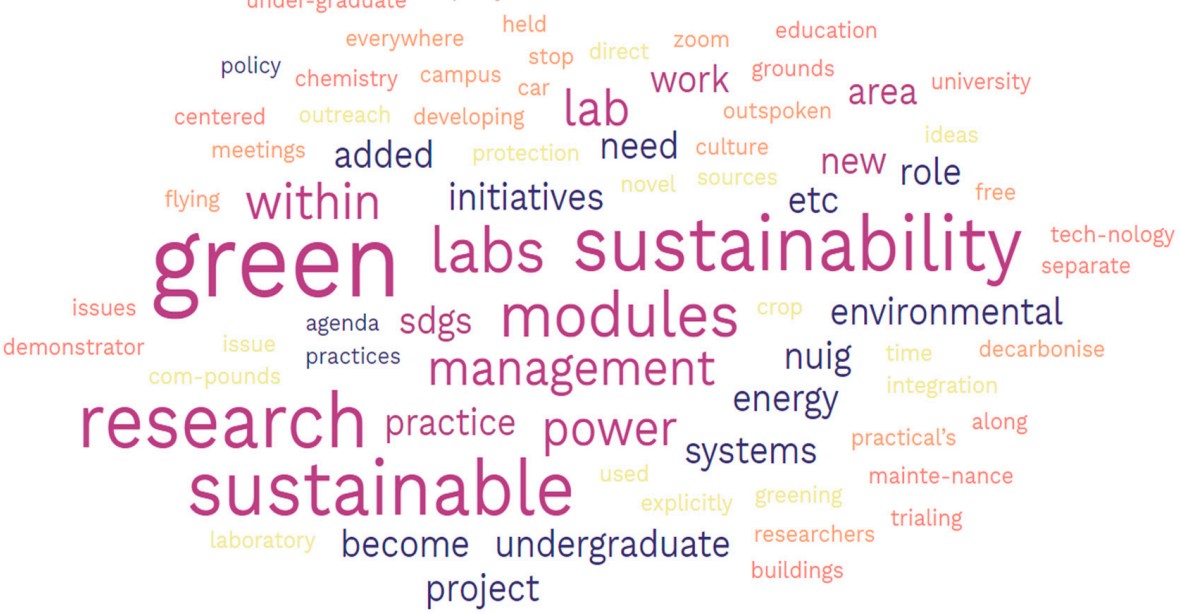

**Figure 5.** Word cloud showing suggestions for furthering sustainable development on campus.

The survey found that 92% of staff were interested in ESD implementation guides and 85% were interested in training or a workshop on how to implement the SDGs into their modules. In total, 76% said they would be interested in having their module(s) analysed by the keyword scan described in this study. The most common barriers to ESD implementation for staff include setting aside the time/resources, lack of official guides, and not knowing where to start, respectively. Staff expressed the desire for an institution-wide ESD implementation plan. They feel a lack of direction from upper management. The desire to incorporate ESD is there among a significant portion of staff, but the institution does not recognise the time and resources that need to be allocated to achieve this. When staff were asked to rate various sustainability issues on campus from 1 to 5, 1 being very important 5 being not at all important, the graph shown in Figure 6 was formed. Energy efficiency and renewable energy were seen as the most important of the options given. However, most staff chose to input their own suggestion by choosing "other". The suggestions gathered can be seen in the word cloud in Figure 7.

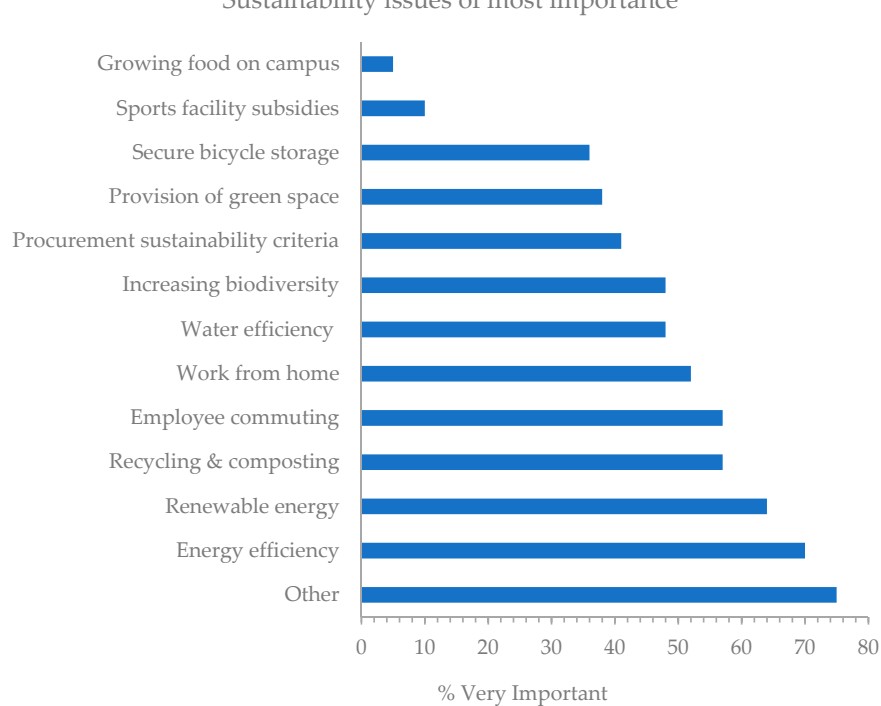

**Figure 6.** Percent of staff that agreed the above are very important sustainability issues on campus.

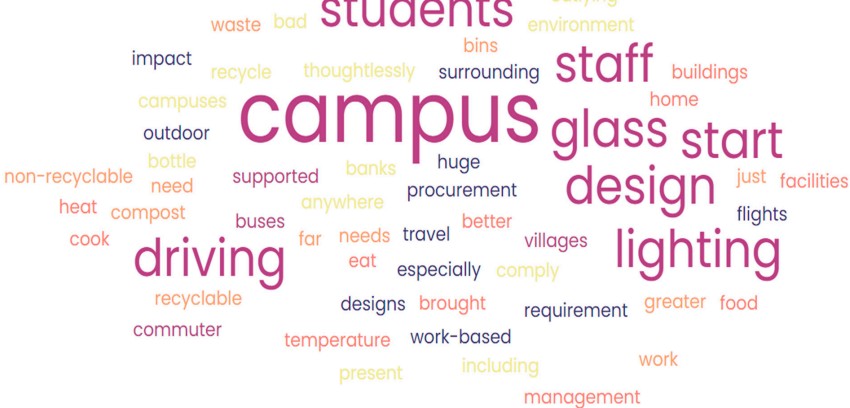

**Figure 7.** Word cloud showing the sustainability issues of most importance according to staff.

### 3.4. Survey vs. Keyword Scan

This section of the results presents a comparison between the two different curriculum mapping methodologies, the keyword scan, and the survey of academic staff. As described in Section 2.4, the survey asked staff to rate their modules in terms of SDG coverage. It asked them to note whether a module covers any of the SDGs in the focused context or in the supportive context, as per the STARS definitions. A total of 33 staff responded to the survey and, of these, 24 staff members rated 91 modules collectively, 3.8 modules per staff member. The survey allowed staff to note if the module covered any of the 1–17 SDGs in either a supportive or focused context. This gave quantitative results which are comparable to the unique number of modules metric or the percent of sustainability in the curriculum, as shown in Table 10 for the un-analysed word lists and Table 16 for the analysed lists. The crucial and GPT keyword lists were scanned for the learning outcomes of the 91 modules rated by staff. The Elsevier and EU Mapper words were also scanned for among these 91 modules for comparison with the crucial and GPT lists, as can be seen in Figure 8. Figures 9 and 10 compare the quantitative survey results with the number of unique modules hit by the crucial and GPT keyword lists, showing focused modules and supportive modules, respectively. The un-analysed keyword lists used in the meta-analysis do not have the STARS relevancy labels, so they could only be compared to the sum of focused and supportive modules under each SDG. Table 17 shows the correlation values for all three metrics discussed in Tables 9–11 for comparison, but it is important to note that the number of unique modules hit represents the direct comparison in results.

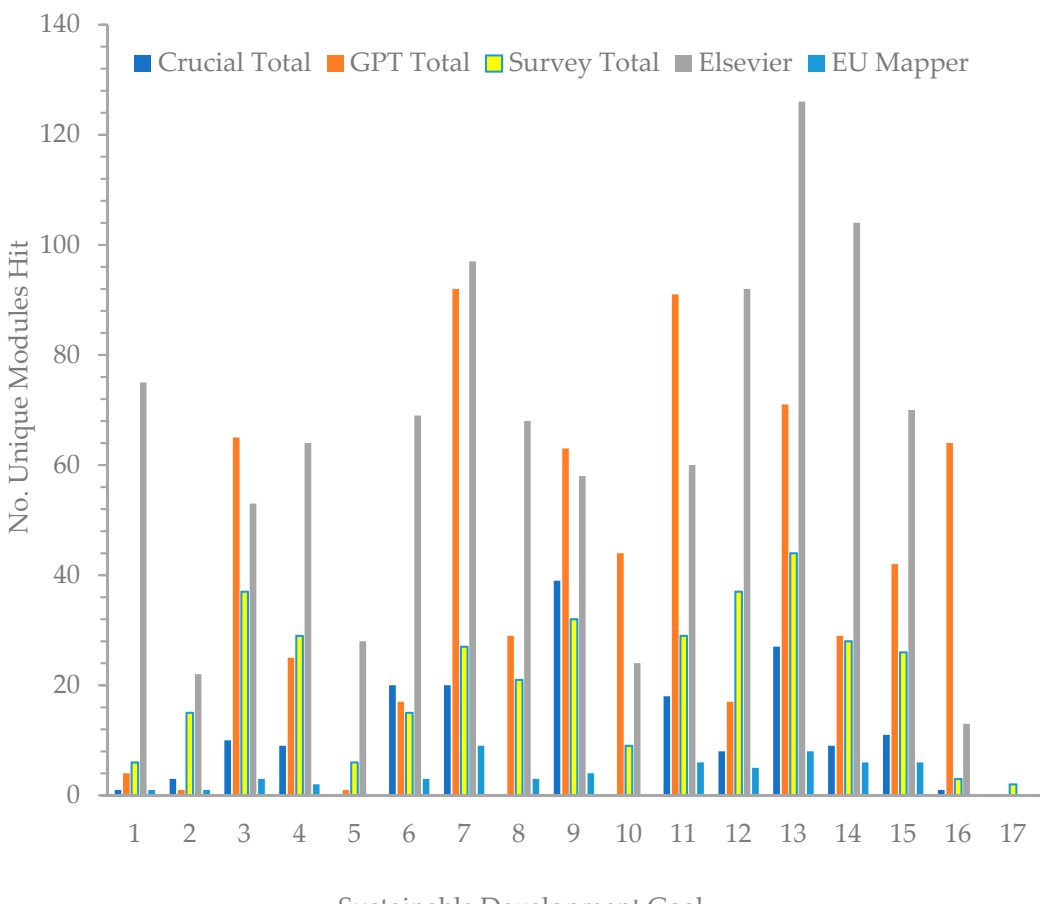

**Figure 8.** Survey versus keyword scan for number of SDG-focused and/or -supportive modules hit.

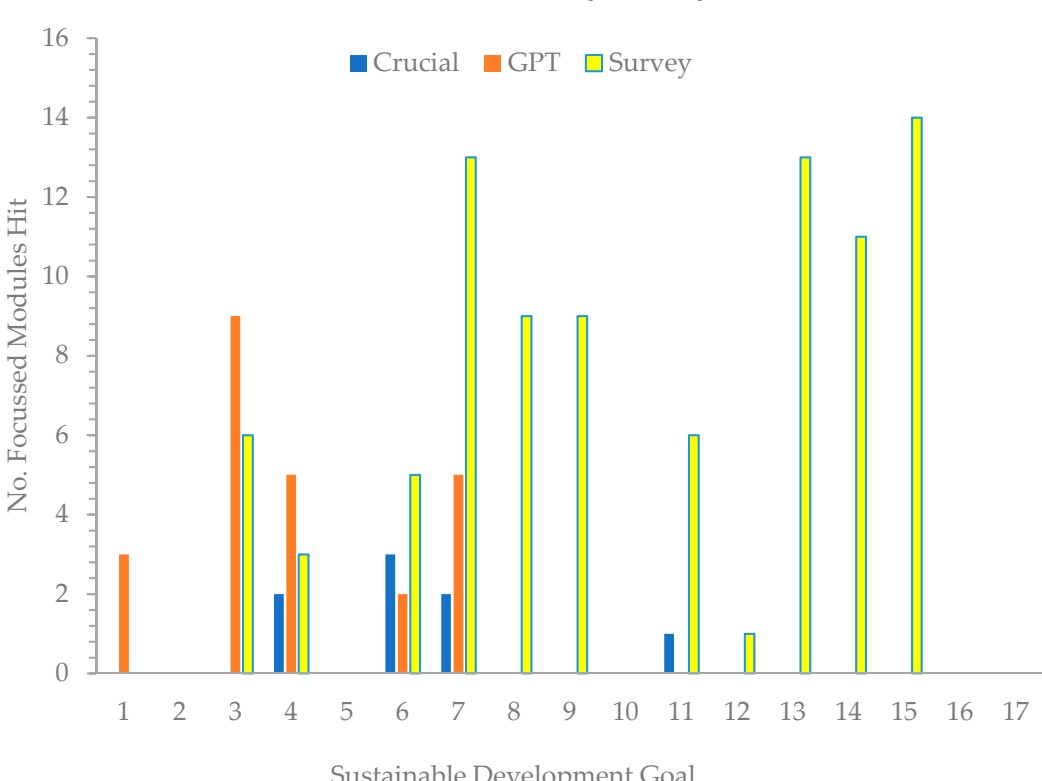

**Figure 9.** Survey versus keyword scan for SDG-focused modules.

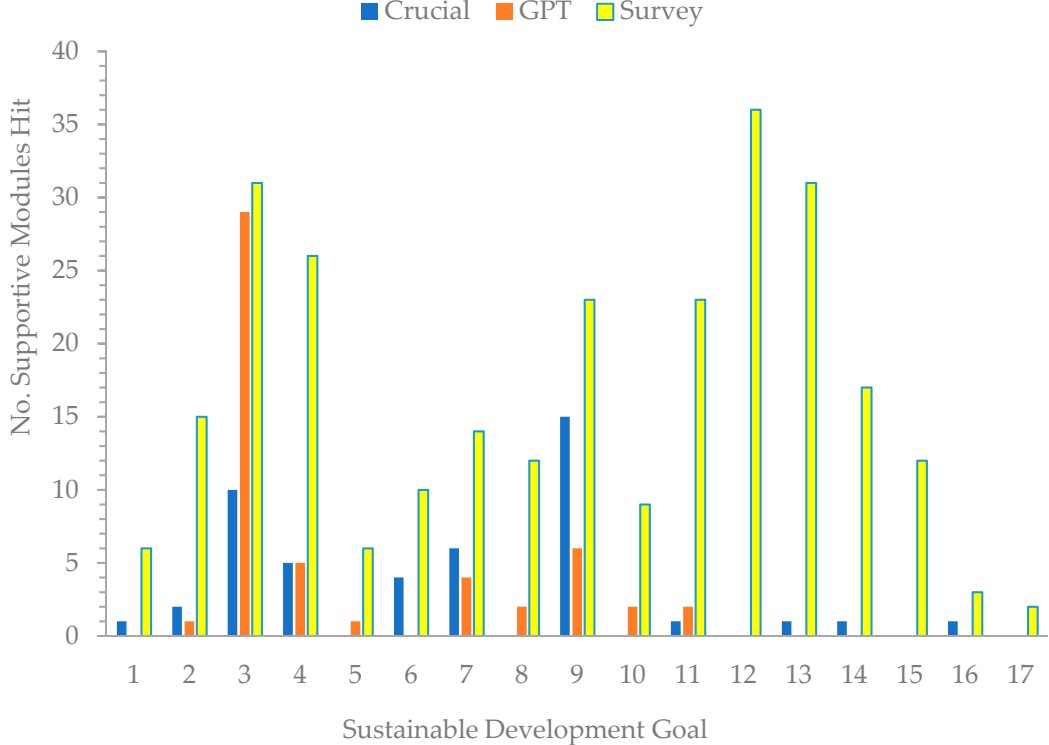

**Figure 10.** Survey versus keyword scan for SDG-supportive modules.

**Table 17.** Correlation coefficients between various keyword lists and the quantitative survey results *.

| Metric | Relevance | Crucial | GPT | EU | Elsevier |
|---|---|---|---|---|---|
| No. Keywords | Focused | 0.24 | 0.4 | | |
| | Supportive | 0.55 | 0.13 | | |
| | Both | 0.74 | 0.5 | 0.48 | 0.57 |
| No. Modules | Focused | −0.09 | 0.28 | | |
| | Supportive | 0.39 | 0.69 | | |
| | Both | 0.36 | 0.64 | 0.12 | −0.45 |
| No. Hits | Focused | 0.26 | 0.57 | | |
| | Supportive | 0.29 | −0.06 | | |
| | Both | 0.54 | 0.44 | 0.59 | 0.69 |

* This table is a heat map, where darker blue aligns with a larger positive number and darker red aligns with a larger negative number.

Given the Elsevier list of keywords was hit in 94% of the 1617 modules, the highest of any list, it was used for comparison to the sum of the number of supportive and focused modules identified by the survey, the crucial keywords, and the GPT keywords. Table 17 shows that there is a moderate negative correlation between the Elsevier scan and the survey results. A negative correlation does not imply any sort of consistency in the data as we are looking for a direct comparison between SDG 1 and SDG 17. The crucial and GPT lists perform better here, with the crucial list showing a moderate positive correlation of 0.36, and GPT achieving a strong positive correlation of 0.64, when looking at the sum of supportive and focused modules. Using this sum figure is useful because we are dealing with a small sample set of just 91 modules. Breaking it into focused and supportive, then, we see the effect of a very small sample size. The supportive modules show the same correlation as the sum as there are more supportive modules than focused. The focused modules have a very weak negative correlation with the crucial words and a weak positive correlation with the GPT words. These numbers represent the similarity in the overall SDG trend achieved by the various methods. We can also look for similarity in the highest rated modules using these methods. Taking the top 10 modules for each keyword list and the survey, the survey has 4 modules in common with the GPT, 3 with the crucial words, and 2 with the Elsevier list. However, when we look at the common modules among the various keyword lists, there is more similarity with the GPT and crucial having 6 modules in common, the crucial and Elsevier lists having 7, and the GPT and Elsevier showing 8 modules in common among the top 10. This consistency in the keyword method shows that there will inevitably be some dissimilarities between the keyword and survey mapping methods. Table 18 shows the module code for the five modules with the most hits by each keyword list and the survey. There is similarity between this table and Table 12, the top modules identified by the un-analysed lists, which include modules ZO417 ("Marine & Coastal Ecology", CE343 ("Sustainable Energy"), and CE514 ("Transportation Systems and Infrastructure II"). These three modules were identified as highly relevant to sustainability, particularly ZO417 which received the most hits in Table 18, for each keyword list and the second most in Table 12. However, this module was not highlighted by the survey as a top module.

**Table 18.** The 5 modules with the most hits for each list of 91 modules analysed in survey [1].

| Crucial | No. Hits | GPT | No. Hits | Survey | No. Hits | Elsevier | No. Hits |
|---|---|---|---|---|---|---|---|
| ZO417 | 19 | ZO417 | 41 | CE464 | 11 | ZO417 | 89 |
| EG400/EG5101 | 13 | BI5108 | 33 | CE6102 | 11 | BI5108 | 51 |
| BI5108 | 13 | EG400/EG5101 | 26 | CE6103 | 11 | EG400/EG5101 | 42 |
| CE343 | 11 | CE514 | 25 | CE344 | 10 | CE514 | 39 |
| CE514 | 11 | CE343 | 24 | CE514 | 10 | CE6103 | 32 |

This table is a heat map, where the darker shades of colour align with high table values. [1] Module information can be seen on University of Galway website [51].

## 4. Discussion

### 4.1. Meta-Analysis

Lists of keywords orientated around the 17 Sustainable Development Goals (SDGs) have been compiled and refined by multiple different research groups using multiple different techniques to compile them. A comparative meta-analysis of multiple keyword lists associated with the SDGs has not yet been published. This paper presents results from such an analysis for five keyword lists that are commonly utilised in SDG-mapping research. The objectives of the meta-analysis are mentioned in Sections 2.6 and 3.1.

Identification of the accuracy of the keyword lists was achieved using two different indicators. The first was a percentage calculated by dividing the total number of unique keywords hit by the total number of keywords for each list. This is useful for noting the performance of the list, as it crosses off keywords if they were hit. The second was a percentage calculated by dividing the total number of modules hit by 1617, the total number of modules scanned. These percentages respectively represent the amount of each list that was useful and the level to which the SDGs are embedded in the curriculum. Two other metrics were analsyed to obtain a better understanding of the performance of each keyword list. The first of these was the frequency of words occurring in certain lists, calculated by dividing the total number of keyword hits by the number of keywords in the list, accounting for words that receive more than one hit. The other was the top performing modules as per each keyword list, to review why these modules performed well and whether there was cross-over between the lists. Finally, an overall trend in SDG coverage for the five lists compiled was also presented. It was possible to plot the SDG trend for the compiled number of keywords under each SDG for comparison, and correlation coefficients were calculated between each list and this compiled trend.

These correlation coefficients or R-values can be seen in Table 8, and they correlate each keyword list with the orange SDG trend seen in Figure 3. These R-values show the similarity in SDG trend between the number of unique keywords hit from each list and the total number of keywords under each SDG when all five lists were compiled. They prove the obvious statement that, as you increase the number of keywords you increase the number of hits. However, what it is showing in Table 8 is that when it comes to the context of module learning outcomes, a list size of over 2000 words certainly seems to be too large, as the three lists that are this large have moderate or strong positive correlations with the number of keywords compiled under each SDG. These lists are those compiled by the University of Auckland, the EU Mapper initiative, and by SIRIS academic. What this correlation means is that when these 2000+ keyword lists were scanned for, the SDG trend found either moderately or strongly mirrors the number of keywords under each SDG. From an accurate keyword list, we would expect an independent trend specific to the curriculum material to present itself, as can be seen with the other two lists, those compiled by the University of Monash and by the Elsevier Research Mapping Initiative. The Monash and Elsevier lists show no correlation to the number of keywords under each SDG, so these lists score above the rest on this metric.

Moving onto the accuracy of each list and looking at Tables 9 and 10, there are two things to take from these results. Firstly, in Table 9, Elsevier and Monash outperform the rest of the lists again with the percent of keywords covered, achieving 36% and 25% coverage, respectively, compared to 11% by Auckland and SIRIS and 4% by the EU Mapper. These figures loosely correlate to the number of words in the list, but not perfectly. From smallest to largest in terms of size, the lists rank as follows: Monash, Elsevier, Auckland, EU Mapper, and SIRIS. Secondly, there is an abnormally high percentage of the curriculum being hit for all lists except for the EU Mapper. The STARS content display can be used to analyse historical data from hundreds of previous reports. The average institution on the system had just 13.3% sustainability module offerings. This display does not show the breakdown of this into sustainability-focused or -supportive, simply an overall percentage of courses offered by the institution which have sustainability-related learning outcomes. Therefore, the scores of 71%, 81%, 88%, and 94% achieved by Auckland, Monash, SIRIS,

and Elsevier, respectively, show a high degree of over-labelling modules as SDG-related. Delving into why the EU Mapper scored so differently here, it appears to be due to the type of keywords this list contains. A "keyword" according to this list is often at least two words and sometimes a full sentence. Granted, only 121 of these 3381 words were hit, but the formatting of these words gives the list a better chance of capturing a realistic percentage of the curriculum.

It was possible to obtain an overall accuracy picture by looking at the top performing modules in the meta-analysis. Table 12 shows the top five modules for each keyword list and Table 13 shows the seven of these which are common to multiple lists. These seven modules have been identified as highly relevant to sustainability by the keyword scan and, when we look at the titles of the modules in Table 13, it shows that this is accurate. Table 14 shows the total number of hits, unique keywords hit, and the SDGs hit for each of these modules. We can see again how using 12,000+ keywords would not be effective, as these modules cover between 14 and 17 SDGs each, a very high estimate that is very unlikely to be realistic. This table also highlights how SDG 5 and SDG 17 are the least covered, even at this high level. The same is mirrored in Table 9 where SDG 17 only had 2% of its keywords hit overall, and SDG 5 only had 8%. This reflects how SDG 5 and SDG 17 contain a particularly ineffective set of words. In Table 10, SDG 17 was omitted because it scored close to 0% coverage in the entire curriculum. However, after SDG 17, we see SDG 6, 12, and 15 all receive just 1% coverage while SDG 2, 5, and 16 receive 2%, a reflection of what SDGs the curriculum covers as opposed to a reflection of a lack of effective words in the keyword list. We can see the SDGs in the curriculum are heavily skewed in favour of SDG 3 and SDG 10. SDG 3 is another example of what happens when we have too many keywords in a category. As can be seen in Table 1, this SDG has almost double the average number of keywords when compared to the rest of the SDGs. SDG 10 can be explained by looking at the last metric to be analysed in the meta-analysis, the frequency of keyword hits, as shown in Table 11. This result highlights where certain lists had outlier words or words receiving an abnormally large amount of hits in the wrong context. As mentioned in Section 3.1, the category of SDG 10 under the Monash list of words shows an obvious outlier which was identified as the keyword "age". This keyword was appearing mid-string in words such as "management" and was skewing the SDG trend in favour of SDG 10. This was very useful going into the critical analysis, where removing words receiving a large amount of hits in the wrong context was an objective.

*4.2. Critical-Analysis and Chat GPT Enrichment*

The keyword critical analysis was successful in that accuracy was dramatically improved. Given the fact that we learned that 2000+ word lists are ineffective in the meta-analysis, the critical analysis shows the other end of the scale. The list of words produced from this section was 222 words long and was split into 66 focused and 166 supportive, giving the opportunity to review how word lists of these smaller sizes compare to those that have upwards of 915 words. Furthermore, the following stage of synonym enrichment by ChatGPT would increase the word count of the critically analysed list, so the critical analysis aimed to reduce the words down to those that crucially had to be there, as they were hit in the correct context of their assigned SDG over 70% of the time.

However, before the critical analysis, words had to be removed through other filters. Over 10,800 words were removed purely due to them never receiving a hit among our 1617 modules. This could be a limitation of the crucial keywords as they are based around a set of meta-data specific to one Irish university. A further 273 words were removed when the list was scanned for duplicates because, as this list was a compilation of the five lists studied in the meta-analysis, there were bound to be some duplications. Then, the analysis began and 732 words were removed due to a high occurrence in the wrong context. Looking at Table 4 in Section 2.7, the analysis started with words hit 1–2 times and words hit 50+ times. The low frequency words would be quick to filter through, as they had a max of two examples to manually review, and the outliers were deemed likely to be mostly

in the wrong context with so many hits. This hypothesis was correct and, after the analysis, 65 low-frequency words remained and just 6 words receiving more than 50 hits were kept. These high-hitting words, in order of frequency from highest to lowest are; technology, sustainable, culture, medical, business, and professional development. Sustainable and professional development are focused words and the others are supportive. Following this review of low- and high-hitting words, the remaining 665 words were scanned for any that could be reduced to root words. An example of a rooted word is "agricultur". Leaving the "e" out at the end means this word could hit a sentence containing "agriculture" or "agricultural development". This meant that any words in the format "agricultural something" could be removed, as these were covered by the root word. Some words, however, needed to be unrooted; for example, the keyword environmental. We can see in Table 5 that this word was an outlier in the meta-analysis and received an enormous 895 hits among the 1617 modules. This word is, of course, important to an SDG keyword list, but it was producing a lot of hits in the wrong context in this format. Therefore, this word had to be unrooted or extended, by adding words such as "environmental impact" and "environmental problem". The rooting process removed a further 404 words.

Finally, the manual group critical analysis took place on the remaining 261 words, where, as described in Section 2.7, members of the university's sustainability team each reviewed a list of learning outcomes which had keyword hits. Each member had to analyse each learning outcome and note whether the keyword was in a supportive, focused, or incorrect context, giving the learning outcome a relevancy score, as shown in Table 6. Each list of learning outcomes that was reviewed had 25% cross-over with other lists, so that there could be a consistency check. This check showed an acceptable level of about 70% consistency among various reviewers' scores and so the process was successful. The group review successfully removed the remaining 39 irrelevant words and labelled all other words as either supportive or focused, as per the STARS definitions. As well as improving the accuracy of the keyword scanning tool, this gave the tool the added capability of indicating whether modules were sustainability supportive or focused, which is necessary when completing a STARS application. With the 222 crucial keywords identified, they were handed over to ChatGPT, as described in Section 2.7. This produced 1081 synonyms of these crucial keywords and, when these synonyms were cross-checked against the 12,000+ words that were removed, there were 118 duplicates, meaning there were 963 unique keywords added by this process. This brings the total number of keywords in the "GPT" list to 1185, in between the 915-word Monash list and 1600-word Elsevier list.

The critical analysis and synonym addition had a significant positive effect on the performance of the keywords. Table 15 shows the percent of keywords hit for each list and compares it to the average value from the meta-analysis. We know from the meta-analysis that the Elsevier list scored the highest here at 36% and that the average was 13%. The GPT list scored 36% also, meaning it contained a higher portion of effective words than the four other lists studied, again a sign that these lists should be around the 900–1600 word mark. The crucial list scored almost double that of the Elsevier list at 69% of its keywords hit. The low word count of the list comes into effect here meaning fewer words had to be hit to score such a high percentage out of 222 words. However, the crucial list shows more total unique keywords hit at 154 when compared to the EU Mapper list at 121, a list containing over 3300 keywords. A limitation to the crucial list can be seen in SDG 2 and SDG 6 where 100% of the keywords have been hit. This indicates the need for more words in these categories. However, this level of coverage shows the critical analysis was successful, as the goal was to identify the most crucial words; therefore, if all words were hit then all must have been crucial. The righthand side of Table 15 shows the frequency with which words were hit under each SDG, something that was used in the meta-analysis for identifying outlier words. We can see that SDG 4 for the crucial list and SDG 9 for both lists had a lot of hits. When we investigate why this is the case for these lists, we find that SDG 4 contains the keywords "culture" and "professional development", and that SDG 9 contains "technology" and "business"; four of the six remaining outliers are in these SDGs.

Three of these are labelled as supportive which, as we see in Table 16, makes a difference to the percent of sustainability in the curriculum.

Table 16 shows the number of unique modules hit by each keyword list divided by 1617 to ascertain the percent of the curriculum that is focused on or supportive of each SDG, according to each list. We see the effect of the relevancy labels here, where many of these high-hitting words have been labelled as supportive due to too many of their hits not being in a focused enough context. This gives a much more realistic picture of the percent of SDG-focused modules, with the crucial list identifying 7% and the GPT list identifying 19%. As previously mentioned, the average institution on the STARS platform had 13.3% sustainability module offerings, which is the average of 7 and 19, suggesting these lists are a degree more accurate than the previous lists, except for the EU Mapper which identified 8% of modules. The issue with the EU Mapper is that the crucial and GPT lists labelled 36% and 65% of the curriculum as sustainability supportive, respectively. These percentages are closer to those achieved by the other lists, which suggests that, if the other lists were split into focused and supportive words, they might achieve a realistic percentage of each, while the EU Mapper is at maximum capacity at 8%, showing it is lacking in the level of coverage it offers.

The critical analysis was successful in identifying crucially significant keywords and the addition of synonyms bulked the list back to a useful size, showing the effectiveness of the combination of the critical analysis and the intelligent text manipulation model. This process utilised artificial intelligence to produce contextually relevant synonyms to enrich critically analysed keywords. This captured the expertise offered by the sustainability reviewing team and enhanced it through intelligent word-mapping.

### 4.3. Survey Questionnaire

The survey's questionnaire section showed that the 33 staff members who responded to it already had a strong interest in sustainability, as highlighted in Section 3.3. There were plenty of suggestions for improving sustainability on campus, as seen in Figure 5. Ten of the nineteen suggestions summarised in Figure 5 were in relation to the better teaching of sustainability on campus, with a particular focus on what happens in labs, with the goal of greening labs. Figure 6 shows how staff rated issues of importance and Figure 7 shows other issues of importance that staff put forward. They agreed that the university should be developing the sustainability literacy of its students and 70% of respondents showed interest in joining a sustainability teaching group. What this shows is that there is a cohort of staff who are independently thinking about sustainability in their daily lives and in their careers on campus. There is a clear appetite among staff for a chance to contribute towards sustainable development on campus and they recognise the ability to use their role as educators to do so.

When asked what barriers lie before them, preventing them from implementing sustainability topics into their teaching, the responses reflected what has been shown in the literature. Staff would rather a systematic, holistic approach was taken for ESD implementation, rather than it falling to those who will take it up on their own initiative. They want support financially and in guidance for setting the time aside to consider ESD, the SDGs, and their own teaching. There was a strong interest shown in ESD implementation guides, training, or workshops on how to implement the SDGs into modules, and in an automated analysis of their modules by the keyword tool described in this study. Research has shown repeatedly that a holistic, whole-of-institution approach is the most favorable and effective method for successful ESD implementation.

### 4.4. Survey vs. Keyword Scan

Twenty-four out of the thirty-three staff members that responded to the survey submitted module ratings, and they submitted 3.8 modules each on average, totaling 91 unique modules. Of the 91 modules, 51% were given both focused and supportive ratings, 37% just supportive, 8% just focused, and 4% were given no rating. In other words, 58% of the

submitted modules were deemed sustainability-focused and 88% sustainability-supportive. All 17 SDGs are hit in the 80 supportive modules and, in the 53 focused modules, the only SDGs that were omitted were SDG 1, 2, 5, 10, 16, and 17. This is quite an elevated level of SDG coverage because, as previously mentioned, on average, institutions have just 13.3% sustainability modules. This means that, in comparison to the average STARS-rated institution, the staff that responded to this survey overestimated the level to which the SDGs are embedded in their modules.

This is reflected in Figure 8 where the trend in SDG coverage found in the survey is graphed alongside four keyword list trends. The keyword list trends were acquired by scanning for these keywords in the learning outcomes of the 91 modules rated by staff, and the lists used for comparison were the Elsevier, EU Mapper, crucial, and GPT lists. Elsevier was chosen due to the elevated level of accuracy and frequency shown by these keywords in the meta-analysis. The EU mapper words were chosen due to it finding the closest realistic percentage of sustainability in the curriculum of the un-analysed lists. The crucial and GPT lists were compared to analyse their accuracy and performance. Table 17 shows the correlation coefficients for the data plotted in Figure 8 and for other metrics of the same dataset. The number of unique modules hit is plotted in Figure 8, as this is the metric that the survey collected. Taking the number of unique modules hit first, we can see the EU mapper has no correlation ($r = 0.12$) and Elsevier has a negative correlation ($r = -0.45$) which effectively has no correlation. The crucial set of words have a moderate correlation ($r = 0.36$) and the GPT words have a strong correlation ($r = 0.64$). This is an incredibly positive result for the crucial and GPT word lists, proving they can highlight the SDGs that the module owners consider their modules relevant to with a moderate/strong degree of accuracy, while the un-analysed lists could not.

Figures 9 and 10 break the results into focused modules and supportive modules, meaning only the crucial and GPT word lists could be compared. The high frequency of survey-rated modules can be seen here with the survey coverage of many more modules and SDGs. In terms of focused SDG hits, the survey found a total of 90, while the GPT found 24, and crucial found 8. In terms of supportive modules, the survey found 276, the GPT 52, and crucial 47. The correlation on the focused side was none ($r = -0.09$) for the crucial words and weak ($r = 0.28$) for the GPT. The correlation on the supportive modules was moderate ($r = 0.39$) for the crucial words and strong ($r = 0.69$) for the GPT. The higher level of correlation seen in the supportive hits could be due to the higher sample size as well as the fact that there are more supportive keywords. However, a real correlation has been found here, and this shows the GPT list of keywords has had the best success in capturing a realistic picture of where the SDGs are embedded in the curriculum and of what staff members themselves think. The GPT list identified 26% focused modules and 58% supportive among the 91 modules submitted by staff. As mentioned, this cohort of staff were already invested in sustainability, so it is likely that their modules have a higher level of SDG content than the average across the institution.

## 5. Conclusions

This research successfully presented two methodologies for measuring a baseline of SDG teaching in a university's curriculum, the keyword scan and the survey. The meta-analysis showed the variations that can occur in keyword scans by using different keyword lists, highlighting the importance of reviewing the results from such keyword scans. The critical analysis and group review of keywords was successful in increasing the accuracy of the crucial keyword list and the automated enrichment by artificial intelligence increased the accuracy further, through the additions of contextually relevant synonyms to the crucial list. It was also shown that the keyword method captures a more realistic picture of where the SDGs lie, as this method tended to be closer to the average level of sustainability modules found among STARS reports, this being 13.3%. The survey produced an SDG trend that was strongly consistent with the keyword tool when using the crucial list; furthermore, it has ChatGPT synonyms, showing the keyword scan can provide results

that agree with the input from module owners to a substantial extent. The barriers for staff to implement ESD were found to be top-down support, with staff expressing an inability to find the time or resources to carry out an SDG analysis and consider implementing ESD. Staff expressed a desire for ESD implementation guides, training, and tools, as well as for a holistic, whole-of-institution implementation approach.

**Supplementary Materials:** The following supporting information can be downloaded at: https://www.universityofgalway.ie/sustainability/learn-live-lead-model/learn/teachinglearning/ (accessed on 2 April 2023), Template SDG Keyword Scanner.

**Author Contributions:** Conceptualisation, methodology, J.G. and T.A.; software, S.M.J. and T.A.; validation, J.G.; formal analysis, T.A.; investigation, T.A.; resources, T.A.; data curation, T.A.; writing—original draft preparation, T.A.; writing—review and editing, J.G.; visualisation, T.A.; supervision, J.G.; project administration, J.G.; funding acquisition, J.G. All authors have read and agreed to the published version of the manuscript.

**Funding:** This research was funded by the Science Foundation Ireland (SFI) through the MaREI Centre (Grant no. 12/RC/2302_2). The authors also acknowledge the financial support from Enterprise Ireland for Construct Innovate, Ireland's National Research Centre for Construction Technology and Innovation, under grant agreement no. TC-2022-0033.

**Institutional Review Board Statement:** Not applicable.

**Informed Consent Statement:** Not applicable.

**Data Availability Statement:** Data supporting the reported results can be obtained by contacting the authors.

**Acknowledgments:** Learning outcome data were provided by the University of Galway. Thanks to members of the CUSP academic working group, who contributed to the critical analysis of the keywords, and academic staff in the College of Science and Engineering who participated in the survey. The survey template was provided by Philippe Lemarchand of Technological University Dublin.

**Conflicts of Interest:** The authors declare no conflict of interest. The funders had no role in the study's design; in the collection, analyses, or interpretation of data; in the writing of the manuscript; or in the decision to publish the results.

**Appendix A**

Sustainability-focused learning outcomes are student learning outcomes that explicitly address the concept of sustainability. A learning outcome does not necessarily have to include the term "sustainability" to count as sustainability-focused, as long as there is an explicit focus on the interdependence of ecological systems and social/economic systems. Specific examples include (but are not limited to):

- Students will be able to define sustainability and identify major sustainability challenges.
- Students will understand the carrying capacity of ecosystems related to providing for human needs.
- Students will be able to apply concepts of sustainable development to address sustainability challenges in a global context.
- Students will identify, act on, and evaluate their professional and personal actions with a knowledge and appreciation of interconnections among economic, environmental, and social perspectives.

Sustainability-supportive learning outcomes are student learning outcomes that include specific intellectual and practical skills (and/or attitudes and values) that are critical for addressing sustainability challenges, but do not explicitly address the concept of sustainability (e.g., systems and holistic thinking, change agent skills, interdisciplinary capacities, social and ethical responsibility). Specific examples include (but are not limited to):

- Students will be able to demonstrate an understanding of the nature of systems.
- Students will understand their social responsibility as future professionals and citizens.

- Students will be able to accommodate individual differences in their decisions and actions and negotiate across them.
- Students will analyse power, inequality structures, and social systems that govern individual and communal life.
- Students will be able to recognise the global implications of their actions.

**Appendix B**

Pseudocode/Algorithm Steps for Matching Tool:
Step 1: Reading data from excel to data frame
data = read_excel(file_path)
Step 2: Preprocessing words to obtain SDGall, sdg, and relevancy label to list of records

- This step involves applying a series of text preprocessing operations to the raw data, which can be represented as a series of mathematical functions or operations. For example, we might apply the following functions:
- To convert text to lowercase: f(text) = lowercase(text)
- To remove punctuation: g(text) = remove_punctuation(text)
- To tokenise the text into individual words: h(text) = tokenise(text)
- To remove stop words (common words that do not carry much meaning): i(words) = remove_stopwords(words)
- These functions can be combined in various ways to create a preprocessing pipeline, such as:
- preprocess(text) = i(h(g(f(text))))
- The resulting preprocessed data can be represented as a set of records, where each record contains the preprocessed text for SDGall, sdg, and relevancy.

Step 3: Using the list of records to find words matching the learning outcomes

- This step involves comparing the preprocessed text from Step 2 against a set of learning outcomes. This can be represented as a series of functions or operations:
- To match words from a list against a piece of text: j(text, words) = count_matching_words(text, words) > 0
- To find all records that match at least one learning outcome: k(records, outcomes) = [r for r in records if j(r['SDGall'], outcomes)]
- The result of this step is a list of records that match at least one learning outcome.

Step 4: Creating a new file with the matched records

- This step involves writing the matched records from Step 3 to a new CSV file. This can be represented as a single function or operation:
- To write a list of records to a CSV file: write_csv(records, filename)
- The resulting file will contain the data from the matched records, along with any associated metadata (such as relevancy labels).

Python Code Example:

```
// Step 1: Reading data from Excel to DataFrame
data = read_excel(file_path)
// Step 2: Preprocessing to extract relevant fields
records = []
for row in data:
    record = {
        'sdgall': preprocess(row['sdgall']),
        'sdg': preprocess(row['sdg']),
        'relevancy': preprocess(row['relevancy'])
    }
    records.append(record)
// Step 3: Finding words matching learning outcomes
matched_records = []
```

```
for record in records:
    match_status = False
    for outcome in curriculum_metadata:
        if match_words(record['sdgall'], outcome['learning_outcomes']):
            match_status = True
            break
    if match_status:
        matched_records.append(record)
// Step 4: Creating new file with matched records
create_csv(matched_records, 'all_sdgs.csv')

Python Full Code:
import pandas as pd

# Read the Curriculum Metadata file
df_curriculum = pd.read_excel("Curriculum Metadata.xlsx")

# Convert the "Learning Outcomes" column to string format
df_curriculum["Learning Outcomes"] = df_curriculum["Learning Outcomes"].astype(str)

# Read the SDG Keyword Database file
df_sdg = pd.read_excel("SDG Keyword Database.xlsx")

# Convert the "SDGALL" column to string format
df_sdg["SDGALL"] = df_sdg["SDGALL"].astype(str)

def preprocess(df):
    records = df[['SDGALL', 'SDG', 'Relevancy Label']].to_records(index=False).tolist()
    return [(str(word), str(sdg), str(label)) for word, sdg, label in records]
def replace_nan_in_dataframe(df):
    df = df.fillna("")
    return df

def find_words_matching(df2, records):
    requiredfields = ['Module Title (Quercus)', 'Module Code (Quercus)', 'ECTS', 'Year',
                'Stream', 'Instance', 'Course Description', 'Course Stream Desc',
                'Grad Level', 'Course', 'Course Level', 'Taught Research', 'Faculty',
                'Department', 'Module Code Akari', 'Module Title Akari', 'Long Title',
                ' Start Date', 'End Date', 'Status', 'Owner', 'Credits',
                ' Level', 'Version', 'EQF Level', 'Date Approved', 'Courses']
    alldictionaries = []
    for i in records:
        print("Searching for word '{}' ... ".format(i[0]))
        status = df2['Learning Outcomes'].apply(lambda x: i[0].lower() in x.lower())
        if True in list(status):
            rows = df2.loc[status]
            for index, row in rows.iterrows():
                dictrecord = row[requiredfields].to_dict()
                dictrecord['keyword'] = i[0]
                dictrecord['Learning Outcome'] = row['Learning Outcomes']
                dictrecord['SDG'] = i[1]
                dictrecord['Relevancy Label'] = i[2]
                alldictionaries.append(dictrecord)
            print("Added record(s)")
        else:
```

```
                    print("No match found for this word!")
                    resultant = pd.DataFrame(alldictionaries)
                    return resultant

            if __name__=="__main__":
                    df = pd.read_excel("SDG Keyword Database.xlsx")
                    df2 = pd.read_excel("Curriculum Metadata.xlsx")
                    df2 = replace_nan_in_dataframe(df2)
                    words = preprocess(df)
                    resultantdf = find_words_matching(df2,words)
                    resultantdf.to_csv("all_sdgs.csv")
```

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
