# Peer review of "Education for Sustainable Development: Mapping the SDGs to University Curricula"

_sustainability, doi:10.3390/su15108340_

Round 1

Reviewer 1 Report

Line 92 and 226: “Technological University of Dublin” instead of “Technical University of Dublin”

Line 107: I think the point “2.1 Sustainability Assessment Tools for HEIs” should go in the introduction but not in Material and Methods.

Line 196-197: “we extract the values of the SDGall”… I wonder if you can write “we extract the values of all the SDG (SDGall)” or something like that.

In the manuscript, sometimes the word " SDGall " is written as "SDGall" and sometimes as "sdgall".

Line 280: Please, review in Table 2 the following: SDG 1-171. The last “1” should be a superscript.

Line 296: Please add a full stop after the word “lists”  

Line 315: Suggestion “In the first stage, words”

Line 336: In table 5, perhaps you should write “environment” instead of “Environment”. And perhaps “Misc.” instead of “Misc”

Line 405-406: I wonder if it is “The trends in comparison are for the following four metrics; the total number of hits,…” or “The trends in comparison are for the following four metrics: the total number of hits,…”. The same in line 429.

Line 426: Please, in Figure 3, you should add tick marks in the y axe. The same for Figure 4, Figure 6 (x axe), Figure 8, 9 and 10.

Line 485: I wonder if you can add a note in Table 12 and Table 18 with the webpage where courses can be found. https://www.universityofgalway.ie/course-information/module/#

Lines 881-886: I wonder if it is necessary to rewrite the research questions that are already mentioned in lines 94-98.

Author Response

Line 92 and 226: “Technological University of Dublin” instead of “Technical University of Dublin”

  • Completed: Fixed spelling.

Line 107: I think the point “2.1 Sustainability Assessment Tools for HEIs” should go in the introduction but not in Material and Methods.

  • Completed: Moved section 2.1 to the introduction, line 88.

Line 196-197: “we extract the values of the SDGall”… I wonder if you can write “we extract the values of all the SDG (SDGall)” or something like that.

  • Completed: The line now reads “For each row in the DataFrame, we extract the values of the keywords (SDGall column), the SDG, and relevancy columns, and preprocess them using a series of text processing steps.”

In the manuscript, sometimes the word " SDGall " is written as "SDGall" and sometimes as "sdgall".

  • Completed: Changed all to SDGall.

Line 280: Please, review in Table 2 the following: SDG 1-171. The last “1” should be a superscript.

  • Completed: Changed to superscript

Line 296: Please add a full stop after the word “lists” 

  • Completed: Full stop added.

Line 315: Suggestion “In the first stage, words”

  • Completed: Comma added.

Line 336: In table 5, perhaps you should write “environment” instead of “Environment”. And perhaps “Misc.” instead of “Misc”

  • Completed: Now reads "environment" and "Misc."

Line 405-406: I wonder if it is “The trends in comparison are for the following four metrics; the total number of hits,…” or “The trends in comparison are for the following four metrics: the total number of hits,…”. The same in line 429.

  • Completed: Colon is more appropriate.

Line 426: Please, in Figure 3, you should add tick marks in the y axe. The same for Figure 4, Figure 6 (x axe), Figure 8, 9 and 10.

  • Completed: tick marks added to Figures 4, 6, 8 ,9 and 10.

Line 485: I wonder if you can add a note in Table 12 and Table 18 with the webpage where courses can be found. https://www.universityofgalway.ie/course-information/module/

  • Completed: Reference to University of Galway website added under Table 12 and Table 18.

Lines 881-886: I wonder if it is necessary to rewrite the research questions that are already mentioned in lines 94-98.

  • Completed: First 3 lines of conclusion removed.

Reviewer 2 Report

This paper describes a mapping criteria of the SDGs to university curricula to realize sustainable development in an effective manner. The paper discusses some important issues and ground realities for the implementation of sustainability on campus. Moreover, it is suggested that the keyword method captures a more realistic picture about where the SDG lie in the reports. 

Although the present research is focused and well designed, following observations are found after reviewing the manuscript:

The description of the previous work connecting the present research looks awkward in the abstract. For example:

This paper expands on previous research titled "Embedment of the Sustainable Devel- 12

opment Goals in Engineering Degree Programmes", in which a keyword scanning tool was created to quantify the level of SDG coverage within a list of learning outcomes ...

This should be discussed in the introduction part. In the abstract, the focus should be on the novelty introduced in the current work.

Fig.3 SDG trend for the total 12,486 keywords showing various metrics are depicted for SDG 1-16. What about the missing SDG 17? 

Same in Table 10.

Fig. 6 'Sustainnability issues of most importance', correction required for Sustainability spelling

Several typos in the text that must be corrected in the revised version. For example:

Table 2, Line 1 SDG covered should be SDG 1-17 not SDG 1-171.

Line 289: 'the next stage of the project, critical analysis. The critical analysis ...' should be modified as:

'the next stage of the project, i.e. the critical analysis which would compile ...'

Please request revision.

Author Response

The description of the previous work connecting the present research looks awkward in the abstract. For example: This paper expands on previous research titled "Embedment of the Sustainable Development Goals in Engineering Degree Programmes", in which a keyword scanning tool was created to quantify the level of SDG coverage within a list of learning outcomes ... This should be discussed in the introduction part. In the abstract, the focus should be on the novelty introduced in the current work.

  • Completed: Reworded the first 5 sentences of the abstract as follows: Education for sustainable development (ESD) is a growing research field, particularly over the last decade. Measuring the level of ESD that is educated currently is useful for planning further implementation of sustainability related teaching. The SDGs are a useful benchmark for sustainability topics and so this paper follows a methodology in which a keyword scanning tool was created to quantify the level of SDG coverage within a list of learning outcomes. The aim of the research is to further develop this methodology and compare results from the keyword tool with results from a survey of module lecturers. SDG related keyword lists were collected from multiple sources for a meta-analysis, examining the performance of various lists.

Fig.3 SDG trend for the total 12,486 keywords showing various metrics are depicted for SDG 1-16. What about the missing SDG 17? Same in Table 10.

  • Completed: SDG 17 is omitted in from certain results (Figure 3, Table 10, 15, 16) due to either SDG 17 receiving 0 hits or receiving such few hits that it wasn’t observable on the graph. I’ve added a sentence on lines 433, 462 and 522 explaining this.

Fig. 6 'Sustainnability issues of most importance', correction required for Sustainability spelling.

  • Completed correct spelling.

Several typos in the text that must be corrected in the revised version. For example: Table 2, Line 1 SDG covered should be SDG 1-17 not SDG 1-171.

  • Completed typos.

Line 289: 'the next stage of the project, critical analysis. The critical analysis ...' should be modified as: 'the next stage of the project, i.e. the critical analysis which would compile ...'

  • Completed with suggested edit.